# Convergent direct and indirect cortical streams shape avoidance decisions in mice via the midline thalamus

Jun Ma[1,2,3], John J. O'Malley[1,3], Malaz Kreiker [1], Yan Leng[1], Isbah Khan[1], Morgan Kindel[1] & Mario A. Penzo [1] ✉

Current concepts of corticothalamic organization in the mammalian brain are mainly based on sensory systems, with less focus on circuits for higher-order cognitive functions. In sensory systems, first-order thalamic relays are driven by subcortical inputs and modulated by cortical feedback, while higher-order relays receive strong excitatory cortical inputs. The applicability of these principles beyond sensory systems is uncertain. We investigated mouse prefronto-thalamic projections to the midline thalamus, revealing distinct top-down control. Unlike sensory systems, this pathway relies on indirect modulation via the thalamic reticular nucleus (TRN). Specifically, the prelimbic area, which influences emotional and motivated behaviors, impacts instrumental avoidance responses through direct and indirect projections to the paraventricular thalamus. Both pathways promote defensive states, but the indirect pathway via the TRN is essential for organizing avoidance decisions through disinhibition. Our findings highlight intra-thalamic circuit dynamics that integrate cortical cognitive signals and their role in shaping complex behaviors.

Contemporary models of corticothalamic circuit organization have predominantly arisen from research within sensory systems, including studies on sensory perception and attention[1,2]. For instance, in sensory systems, thalamic nuclei are traditionally segregated into first- and higher-order relays based on the source of their driver input[3–6], with first-order relays being driven by the periphery (sensory) and higher-order ones being driven by a strong input from layer 5 of the neocortex (L5)[7]. Despite this notion, whether circuit diagrams built from investigations in sensory systems translate to corticothalamic circuits engaged in more advanced cognitive processes, such as learning, memory, reasoning, and decision-making, remains unclear[5,8,9].

A region of the cortex critically implicated in higher-order cognitive functions is the medial prefrontal cortex (mPFC)[10,11]. Research has shown that the mPFC shapes processes such as decision-making, self-referential thinking, and social cognition, and that activity in this brain region increases during tasks that require individuals to evaluate and integrate information about themselves, others, and the broader context[12,13]. Furthermore, the mPFC is implicated in the regulation of emotions and the processing of reward-related stimuli[14–17], which are essential for adaptive behavior. Its intricate connections with other brain regions, including thalamus, enable it to serve as a hub for coordinating complex cognitive processes and emotional responses, shedding light on its significance in understanding the neural basis of human cognition and behavior[18–20].

Investigations into the functional organization of prefronto-thalamic circuits have revealed their importance in various cognitive processes. For example, studies in rodents have shown that the mPFC recruits the mediodorsal thalamus to enhance cortical connectivity and sustain attention[21,22]. The mPFC also targets limbic thalamic circuitry that guides emotional processing and motivated

[1]Section on the Neural Circuits of Emotion and Motivation, National Institute of Mental Health, Bethesda, MD, USA. [2]Laboratory of Anesthesia and Analgesia Application Technology, Xuzhou Medical University, 221004 Xuzhou, China. [3]These authors contributed equally: Jun Ma, John J. O'Malley. ✉e-mail: mario.penzo@nih.gov

behaviors[20,23–26]. Notably, projections from the prelimbic region of the mPFC (PL) to the paraventricular thalamus (PVT) are believed to drive the selection and execution of adaptive responses to appetitive and aversive stimuli including food seeking and conditioned fear[27–30]. Nonetheless, whether top-down control within this pathway adheres to established corticothalamic organizational principles drawn from sensory systems remains uncertain. Indeed, unlike higher-order thalamic nuclei in sensory systems, the PVT receives relatively sparse input from L5 of the mPFC and instead receives prominent innervation from L6[26,31], an avenue via which the cortex provides feedback to thalamus[8,32]. This marked input imbalance raises a crucial question of how the mPFC, particularly PL, exerts top-down control over this brain region to shape emotional and motivated behaviors.

In this study, we harnessed neuroanatomy, patch-clamp electrophysiology, fiber photometry, as well as ex vivo and in vivo calcium recordings, to unveil that, beyond its acknowledged[26,27], albeit weak, direct PL input, the PVT also receives a heretofore undiscovered strong indirect cortical input that is mediated by the thalamic reticular nucleus (TRN). By employing a behavioral paradigm centered on active avoidance—a goal-oriented defensive behavior intricately dependent on both PL and the PVT—we showcase the pivotal role of these circuit dynamics in regulating the flow of cortically-generated information to the PVT, ultimately shaping the selection and manifestation of goal-oriented actions. These collective findings provide further insights into the functional organization of corticothalamic circuits central to higher brain functions.

## Results

### Functional organization of PL-PVT circuits

To investigate the anatomical organization of PL projections to the PVT, we used transgenic lines in which expression of Cre recombinase is restricted to either layer 5 (L5) or layer 6 (L6) pyramidal neurons of the neocortex[33] (since both layers are thought to innervate the thalamus) and used Cre-dependent adeno-associated viral vectors (AAVs) to specifically label the axonal projections of either lamina of PL (Fig. 1a). Using this approach, we found that PL projections to the PVT arise predominantly from L6, consistent with the literature[26,31] (Fig. 1b, c). Of note, we observed that although projections from L5 of PL to the PVT are scarce (Fig. 1c), these projections are abundant in the TRN, a sheet of inhibitory cells that surrounds and constitutes a major source of inhibition to the dorsal thalamus[34] (Fig. 1c). Furthermore, these projections were largely seen in the anteroventral sector of the TRN (avTRN), which innervates the dorsal midline thalamus including the PVT[35] (Fig. 1c). To corroborate our findings, we performed retrograde labeling experiments to identify the layer distribution of PL cell bodies projecting to either PVT or avTRN (Fig. 1d). Consistent with our anterograde tracing experiments, avTRN-projecting PL neurons encompassed both L5 and L6, whereas PVT-projecting cells were nearly exclusively seen in L6 (Fig. 1e–h). In addition, a significant fraction of those PVT-projecting deep layer cells of PL also projected broadly to the anterior TRN, consistent with prototypical thalamic architecture[34,36]. Lastly, using TRIO rabies-assisted mapping of monosynaptic input (See Methods)[37] we found that PVT neurons are almost exclusively innervated by L6 PL neurons (Supplementary Fig. 1).

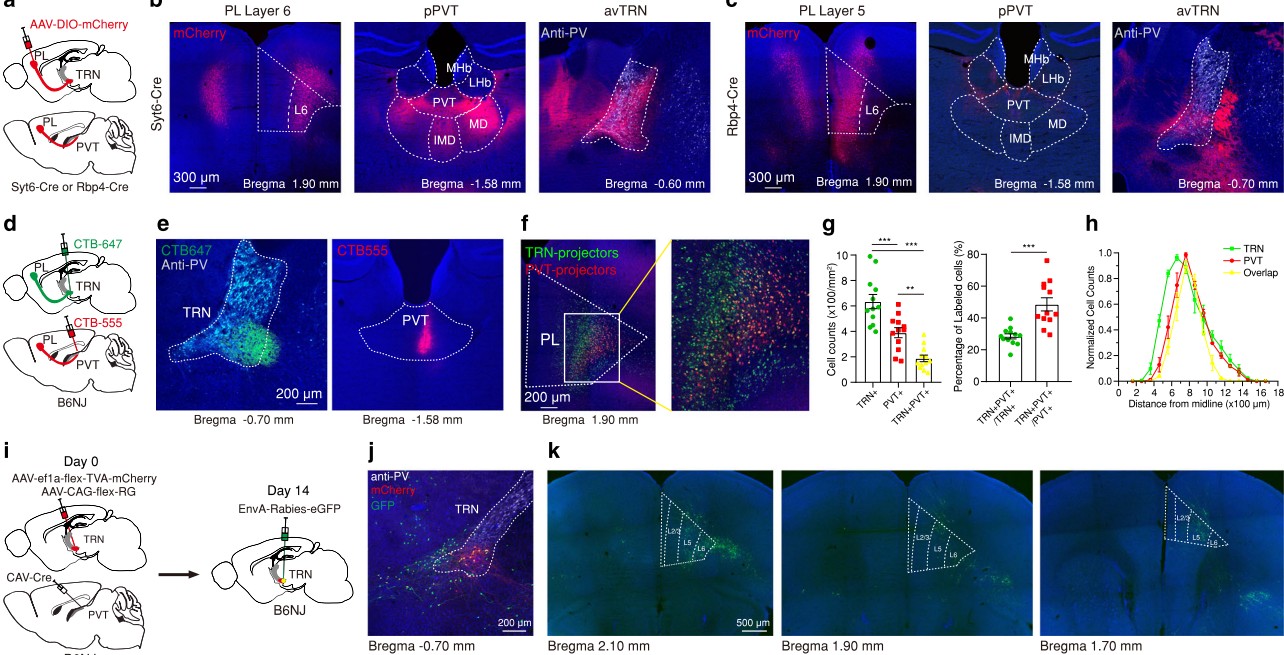

**Fig. 1 | Anatomical distribution of PL projections to the PVT and avTRN. a** Schematic of the viral vector strategy used for anterograde tracing of PL[L5] (Rbp4-Cre) or PL[L6] (Syt6-Cre) projections to the PVT and avTRN. **b** Representative images showing mCherry expression in Layer 6 neurons of the PL (left) and axon terminals within the PVT (middle) and the avTRN (right). **c** Representative images showing mCherry expression in Layer 5 neurons of the PL (left) and axon terminals within the PVT (middle) and the avTRN (right). **d** Schematic of the retrograde tracing strategy used for labeling avTRN- and PVT-projecting PL neurons. **e** Representative images showing the targets for dual-color CTB injections, green into the avTRN (left) and red into the PVT (right). **f**, Representative images showing the retrograde labeled avTRN-projecting cells (green) and PVT-projecting cells (red) are present in the PL region. **g** Left: Quantification of the density of avTRN- and PVT-projecting

neurons in the PL (n = 3 mice, 4 slices per subject) one-way ANOVA, $F_{(2, 33)}$ = 28.15, $P < 0.0001$. Right: Quantification of the percentage of double-projecting cells in the PL. two-tailed t test, $t_{(22)}$ = 4.55, $**P = 0.0046$. **h** Quantification of PL cell counts for each projector, normalized by projector type, plotted along the distance from midline. n = 3 mice. **i** Schematic of the viral vector strategy to trace the PL inputs to PVT−projectors in the avTRN. **j** Representative image showing the rabies starter cells (Rabies-GFP and TVA-mCherry double-labeled cells) in avTRN region. **k** Representative images showing the projectors (Rabies-GFP) of avTRN PVT-projecting cells in the PL. All anatomical experimented were repeated at least once, and similar results were obtained. Data are shown as mean ± s.e.m. Source data are provided as a Source Data file.

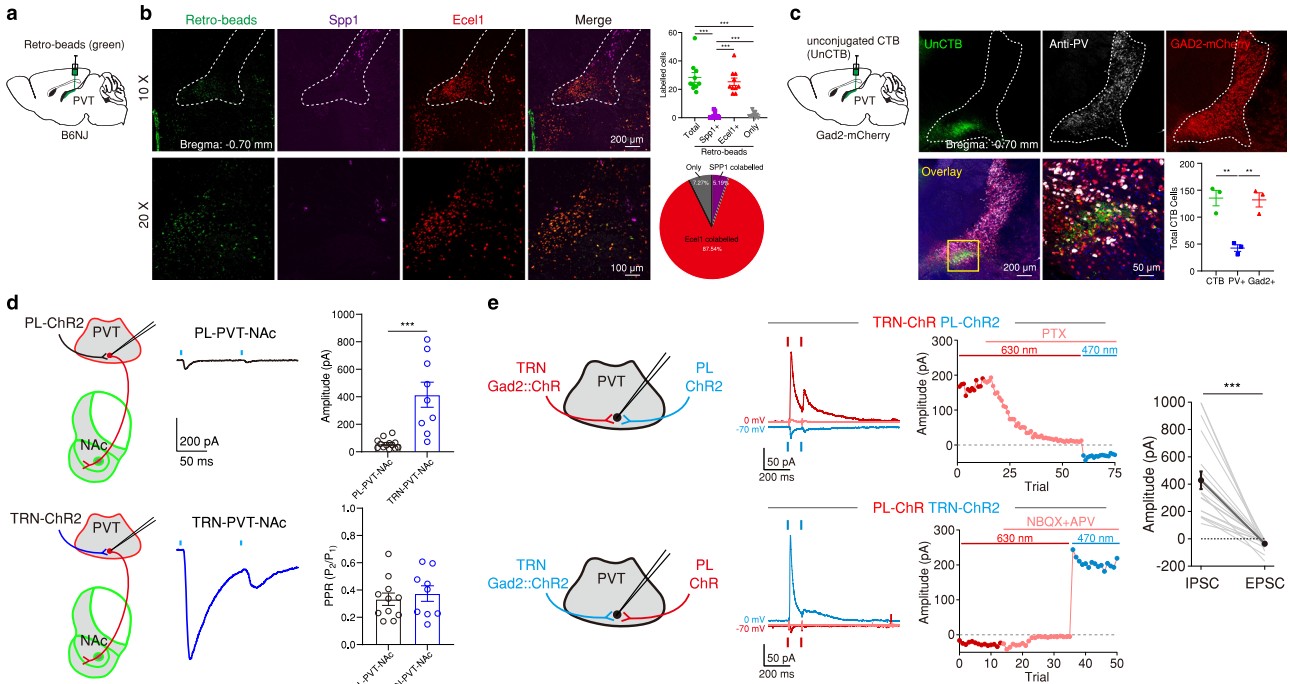

**Fig. 2 | Higher-order sector of the avTRN forms robust monosynaptic connections with NAc-projecting PVT neurons. a** Schematic of the experimental strategy used for molecular identification of PVT-projecting avTRN cells. **b** Top: Representative images of avTRN with retro-beads injected in the pPVT (green) and RNAscope probes for *Spp1* (magenta), *Ecel1* (red) and merged. Bottom: Representative 20x images. Top right: Quantification of avTRN cells labeled with retro-beads, *n* = 10 sections from 3 mice, One-way ANOVA $F_{(3, 27)}$ = 81.16, *P* < 0.0001. Bottom right: Averaged fraction of Retro-beads labeled avTRN cells for each group. **c** Top: Schematic of immunostaining and retrograde labeling of PVT-projecting avTRN cells alongside representative images depicting retrogradely-labeled avTRN cells (unCTB, green), parvalbumin (anti-PV, white), and mCherry (GAD2-mCherry). Bottom left: Overlay of representative images and quantification of total number of avTRN cells labeled with unCTB, PV and GAD2, *n* = 3 mice, 3 slices per mouse, One-

way ANOVA $F_{(2,6)}$ = 19.94, *P* = 0.0022. **d** Left: Schematic of experimental approach for ex vivo slice whole-cell recordings of optically-evoked EPSC (oEPSC, PL) or IPSCs (oIPSC, avTRN) in pPVT-NAc cells. Middle: representative oEPSC and oIPSC traces. Blue lines denote optical stimulation. Right: Quantification of oEPSC and oIPSC amplitudes (independent two-tailed *t* test, $t_{(18)}$ = 4.69, ****P* < 0.0001), and PPRs (independent two-tailed *t* test, $t_{(18)}$ = 0.57), *n* = 11 cells from 4 mice with ChR2 in the PL and 9 cells from 3 mice with ChR2 in the avTRN. **e** Left: Schematic of experimental approach for dual optogenetic recordings of convergent PL and TRN inputs to pPVT cells. Middle two: Representative traces and quantified PSC amplitudes in ACSF and PTX (100 μM) or NBQX (20 μM) + APV (50 μM). Right: Quantification of oIPSC and oEPSC amplitudes, *n* = 19 cells from 9 mice, paired two-tailed *t* test, $t_{(18)}$ = 5.90, *P* < 0.0001. Data are shown as mean ± s.e.m. Source data are provided as a Source Data file.

Together, our findings suggest that the avTRN might be a conduit via which the PL exert top-down control over the PVT, owing to its innervation by L5 input from PL. In partial support of this model, using the TRIO rabies-based monosynaptic tracing method we found that avTRN neurons that project to the PVT, receive input from both L5 and L6 neurons of the PL (Fig. 1i–k), a result which agrees with recent literature[38,39].

Next, we sought to investigate the functional properties of direct and indirect (through TRN) PL input onto PVT neurons, particularly those with efferent projections to the nucleus accumbens (NAc) – a route via which the PVT controls goal-oriented behavior[40–42]. To begin to unravel this question, we first focused on determining the molecular identify of PVT-projecting avTRN neurons. The TRN is divided into two sectors which can be distinguished on the bases of gene expression, anatomical distribution, connectional features, and physiological properties[38,39,43,44]. The 'core' sector of the TRN contains neurons that express genes such as *Spp1* and *Calb1* and project to first-order thalamic nuclei. In contrast, neurons within the TRN 'shell' express *Ecel1* and *Sst* and innervate higher-order nuclei[43,44] (but see ref. 45). Importantly, unlike the core, the shell of the TRN receives input from L5 of the cortex[39], and the strength of the resulting L5-TRN synapses is larger than those of L6-TRN synapses[38]. Using retrograde labeling methods, we found that PVT-projecting cells were largely seen in the 'shell' region of the avTRN and express *Ecel1*[+], consistent with the notion that the PVT participates in higher brain functions (Fig. 2a, b). Notably, however, both retrograde and anterograde labeling demonstrated that

most PVT-projecting avTRN neurons express *Gad2* but not parvalbumin, a prototypical TRN marker[46] (Fig. 2c; Supplementary Fig. 2a, b) (but see ref. 47 for an example of other PV negative cell-types in TRN). Furthermore, all neurons included in our quantifications were in the ventral most portion of the anterior TRN and immediately above the stria medullaris, suggesting that they are more than likely to be within TRN[26,35,41,48,49] and not nearby regions like zona incerta, of lateral hypothalamus which also innervate the PVT[26,50–53]. Next, using patch clamp electrophysiology in acute brain slices in combination with optogenetic stimulation of either PL or avTRN (*Gad2*[+], using *Gad2*-Cre mice) input onto the PVT, we found that while PL-evoked excitatory postsynaptic currents recorded in PVT and more specifically PVT→NAc neurons were typically small, avTRN-evoked inhibitory postsynaptic currents were about an order of magnitude larger (Fig. 2d; Supplementary Fig. 2c, d). Of note, in a separate set of experiments we confirmed that input from both PL and avTRN *Gad2*[+] neurons converge onto individual PVT neurons and display similar differences to those described above (Fig. 2e). Collectively, the corticothalamic architecture illustrated by our findings suggests that PL's influence over the PVT is likely to be a predominantly inhibitory one through the avTRN. Consistent with this prediction, sustained optogenetic stimulation of PL in vivo, produced net inhibitory responses in PVT→NAc neurons and this inhibition was mediated at least in part by the avTRN, since chemogenetic silencing of the avTRN attenuated these responses (Fig. 3; Supplementary Fig. 3). Altogether, our findings suggest that the avTRN might be a conduit via which strong cortical inputs[38,39] exercises top-

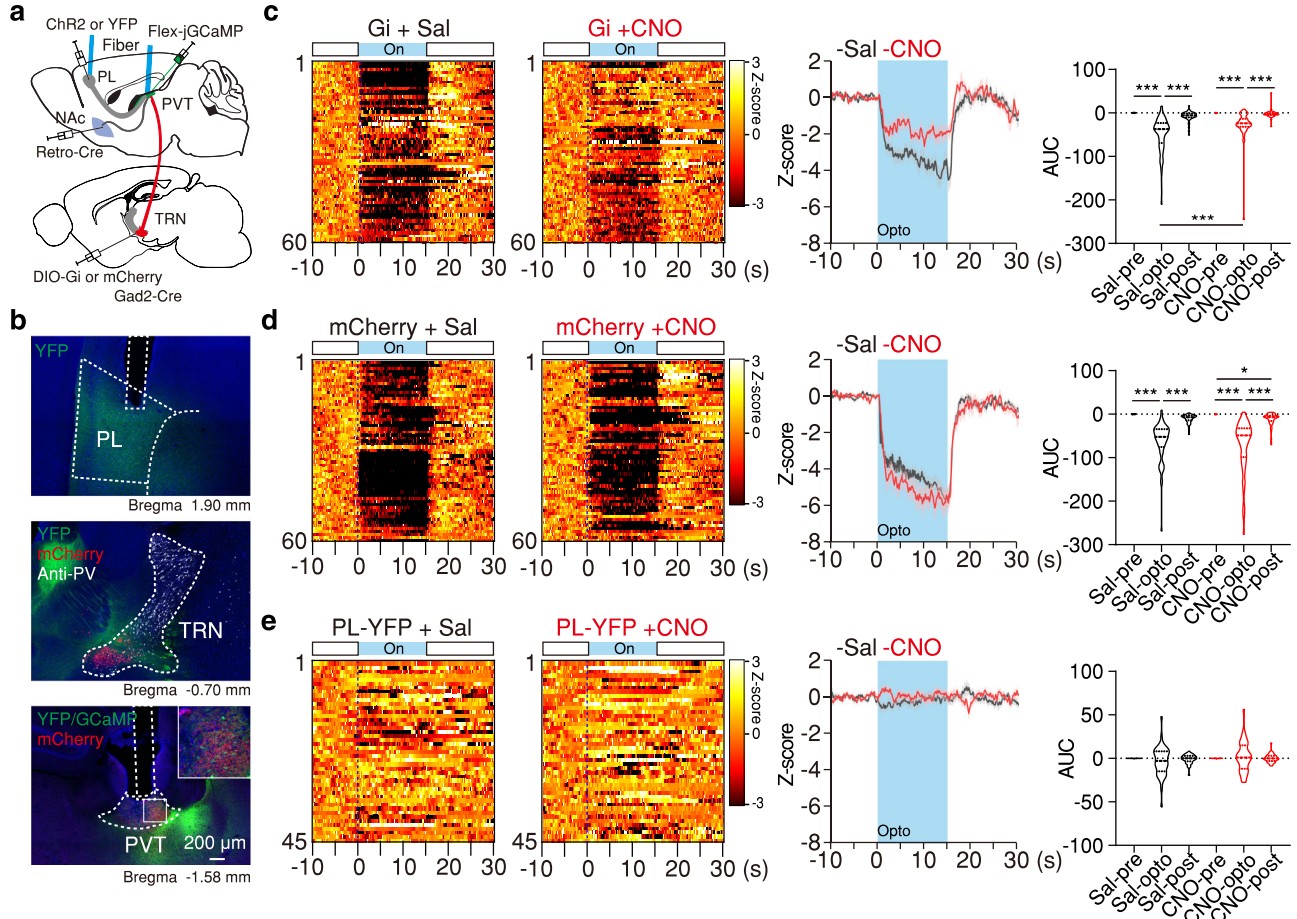

**Fig. 3 | Optogenetic stimulation of the PL drives inhibitory responses in the PVT that are partially rescued by chemogenetic inhibition of avTRN.**
**a** Schematic of the experimental approach for fiber photometry imaging of NAc-projecting PVT cells in response to PL stimulation with and without avTRN inhibition. **b** Representative image of ChR2-YFP expression and optical fiber placement in the PL region (top), ChR2-YFP (green), mCherry-hM4Di (red), and anti-PV (white) expression in TRN region (middle), and GCaMP8s, ChR2-YFP (green), and mCherry-hM4Di (red) expression and fiber placement in pPVT (bottom). **c**–**e** Calcium signal quantification during PL opto-activation and avTRN chemogenetic-inhibition. PL with ChR2-YFP and TRN with mCherry-hM4Di (Gi), $n = 4$ mice (**c**), PL with ChR2-YFP

and TRN with mCherry, $n = 4$ mice (**d**) and PL with YFP and TRN with Gi, $n = 3$ mice. **e** Left: Heatmaps of individual trial calcium responses for Saline and CNO treatment. Middle: Average calcium signal and optogenetic (opto) stimulation duration (blue background). Right: Calcium signal AUC quantification during pre-opto, opto and post-opto, Mixed-effects model (REML) $n = 60$ Trials from 4 mice, $F_{(2, 236)} = 10.65$, $P < 0.0001$. **d**: $n = 60$ Trials from 4 mice, $F_{(2, 236)} = 0.98$, $P = 0.38$. **e**: $n = 45$ Trials from 3 mice, $F_{(2, 176)} = 1.87$, $P = 0.16$. For all quantifications, multiple comparisons were conducted corrected by two-stage linear step-up procedure. Asterisks indicate where ANOVA multiple comparisons found significance. Data are shown as mean ± s.e.m. Source data are provided as a Source Data file.

down control over the midline thalamus to influence behavior. Below, we empirically test this prediction.

## Direct PL-PVT stream promotes defensive states

As noted earlier, the PVT is known to promote goal-oriented behaviors via projections to the NAc[40–42]. This role of the PVT→NAc pathway is thought to be influenced by need-state information arising from the hypothalamus and brainstem and cognitive information arising from the mPFC[54–56]. Recent findings have shown that hypothalamic-derived neuromodulators converge in the PVT to promote goal-oriented behaviors[41,57–59], thus improving our understanding of how need-states are signaled to the midline thalamus. However, the cellular and circuit mechanisms by which cortical signals influence these processes remain unclear. Here, as a proxy for exploring how direct and indirect cortical inputs to the PVT shape goal-oriented actions, we trained mice in the two-way signaled active avoidance behavior task (2AA)[60], since expression of active avoidance behavior depends on an intact PVT→NAc circuit[40] (Fig. 4; Supplementary Fig. 4) (See Methods). In this task, mice must shuttle to the adjacent compartment of a shuttle-box during the presentation of a warning signal (WS) to both terminate the WS and avoid a noxious stimulus (i.e., footshock; FS). Following

training, animals learned to avoid on most trials (Avoidance trials) (Fig. 4b, Supplementary Fig. 4b). As consequence, failure to avoid during the WS was observed in a small proportion of trials in well-trained mice. In these trials, animals shuttled at FS onset—typically within the first second—and the FS was immediately terminated upon shuttling (Escape trials) (Fig. 4b). As such, we categorized trials into either 'Avoidances' or 'Escapes' and independently assessed their associated calcium responses during fiber photometry recordings. Our findings revealed that direct PL → PVT projections were bidirectionally modulated during the 2AA task (Fig. 4b–d; Supplementary Fig. 4). Specifically, while terminals were activated at WS onset for both trial types, i.e., in an outcome independent fashion, (Fig. 4c), the initiation of shuttle responses was associated with a robust suppression of activity in the PL → PVT pathway (Fig. 4d). These response properties contrast with those of PVT→NAc neurons, a pathway where threat representations are outcome dependent (i.e., WS signal responses are selectively observed for avoidance trials) and shuttle responses associated with large calcium transients[40] (Fig. 4g–l). As such, our findings suggest that direct PL → PVT projections may not mediate the differential recruitment of PVT neurons that is associated with avoidance responses. In agreement with our prediction, when inactivating direct

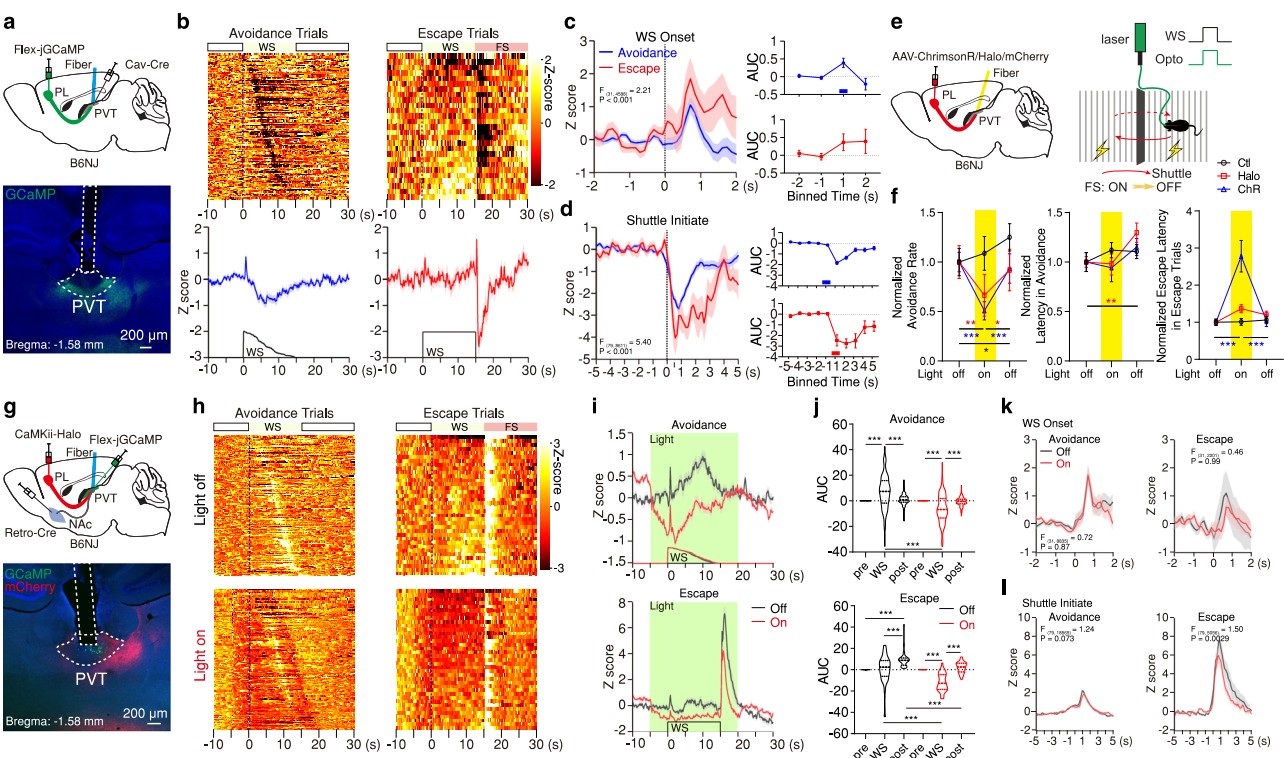

**Fig. 4 | PL–PVT projections are dynamically engaged during active avoidance.**
**a** Top: Schematic of the experimental approach for fiber photometry imaging of PL–PVT terminals. Bottom: Representative image of GCaMP8s expression and fiber placement in pPVT. **b** Heatmaps of individual trial calcium responses for avoidance and escape trials (top), and average signal and WS duration (bottom), $n = 5$ mice. **c–d** Left: Average calcium responses for all avoidance and escape trials during WS onset (**c**) and shuttle initiate events (**d**). Mixed-effects model (REML) interactions reported in graphs. WS onset: Avoidance: $n = 123$ events; Escape: $n = 27$ events. Shuttle initiate: Avoidance: $n = 91$ events; Escape: $n = 20$ events. Right: Calcium signal AUC quantification in 1 s bins. WS onset REML interaction, $F_{(3,444)} = 1.63$, $P = 0.182$; Shuttle Initiate REML interaction $F_{(9,981)} = 6.71$, $P < 0.0001$. **e** Schematic for bidirectional optogenetic manipulations of PL–PVT circuit (left) in the 2AA task (right). **f** Group behavioral data across test sessions normalized to the first light-off session for avoidance rates (left), latencies to avoid (middle), and latencies to escape (right). Ctl: $n = 11$ mice; Halo: $n = 0$ mice; ChR: $n = 10$ mice. Two-way ANOVA. Avoidance rate: $F_{(4,56)} = 5.12$, $P = 0.0014$. Latency to avoid: $F_{(4,53)} = 1.21$, $P = 0.32$. Latency to escape: $F_{(4,56)} = 12.34$, $P < 0.0001$. **g** Schematic of the approach for fiber photometry imaging of NAc-projecting PVT cells and optogenetic silencing of

PL–PVT terminals. Bottom: Representative images of GCaMP8s, Halo-mCherry expression, and fiber placement around pPVT. **h** Heatmaps of calcium responses for avoidance and escape trials in light-off and light-on sessions, $n = 3$ mice. **i** Average calcium signal and WS duration. **j** Calcium signal AUC quantification for avoidance and escape trials, Mixed-effects model (REML) interactions. Avoidance: Off, $n = 144$ Trials; On, $n = 143$ Trials; $F_{(2,570)} = 78.46$, $P < 0.0001$. Escape: Off, $n = 36$ Trials; On, $n = 37$ Trials; $F_{(2,213)} = 14.83$, $P < 0.0001$. **k–l** Averaged calcium responses during WS onset (**k**) and shuttle initiation (**l**) events for all avoidance and escape trials, $n = 3$ mice. Mixed-effects model (REML) interactions shown. WS onset: Avoidance: Off, $n = 144$ Events; On, $n = 143$ Events; Escape: Off, $n = 36$ Events; On, $n = 37$ Events. Shuttle initiate: Avoidance: Off, $n = 117$ Events; On, $n = 120$ Events; Escape: Off, $n = 33$ Events; On, $n = 33$ Events. For all quantifications, multiple comparisons were conducted corrected by a two-stage linear step-up procedure, black lines along the x-axis indicate significant changes reported between groups, and red or blue lines denote the first significant change from the previous bin for within trial type comparisons. Asterisks indicate where ANOVA multiple comparisons found significance. Data are shown as mean ± s.e.m. Source data are provided as a Source Data file.

PL → PVT projections using optogenetics (unilaterally, to prevent behavioral deficits, Supplementary Fig. 5a–e) and simultaneously recording PVT→NAc neurons with fiber photometry, we did not detect any impact of manipulating the direct cortical input on task-related dynamics (e.g., WS, shuttle initiation) in PVT→NAc neurons (Fig. 4g–l; Supplementary Fig. 5f). Importantly, however, this manipulation reliably decreased spontaneous activity within the PVT→NAc pathway (Fig. 4i, j; Supplementary Fig. 5a–d), and bilateral optogenetic silencing of PL → PVT input during the presentation of the WS (but not during the inter-trial interval, ITI) impaired active avoidance (Fig. 4e, f; Supplementary Fig. 6, 7, 8a–c). Interestingly, optogenetic stimulation of PL → PVT projections yielded deficits in both avoidance and freezing to the WS (Fig. 4e, f; Supplementary Fig. 9; Supplementary Fig. 6, 7, 8a–c). Together with prior research[28], these results indicate that intact PL projections to the PVT are necessary for the appropriate execution of defensive responses to learned stimuli, likely by shaping the spontaneous activity of PVT neurons and not by signaling task-related events. Indeed, increases in the spontaneous activity of PVT neurons have been linked to alert states[28,61,62]. Our data supports the notion that

direct PL input to the PVT mediates this process and facilitate threat memory retrieval[29].

## Indirect PL- PVT stream shapes avoidance decisions
Given the lack of influence of direct PL afferents on task-related dynamics within PVT neurons as well as the notion that the mPFC is a major source of top-down control of dorsal midline and medial thalamic nuclei[30,63], we reasoned that the more robust TRN-mediated PL → PVT projections (indirect pathway) we described above may subserve this role. To test this prediction, we first used fiber photometry to monitor the activity of PL cells projecting to the avTRN during the 2AA task. In trained mice, we found that like PL → PVT projections, PL→avTRN cells were bidirectionally modulated during the avoidance task (Fig. 5a–d; Supplementary Fig. 10). However, unlike PL → PVT projections, threat associations in the PL→avTRN pathway appeared to be outcome dependent, with WS onset-related responses observed for avoidance trials only (Fig. 5c). Because a significant proportion of L6 avTRN-projecting PL neurons appear to send collaterals to PVT (Fig. 1), and PL → PVT projections do not display outcome-dependent WS

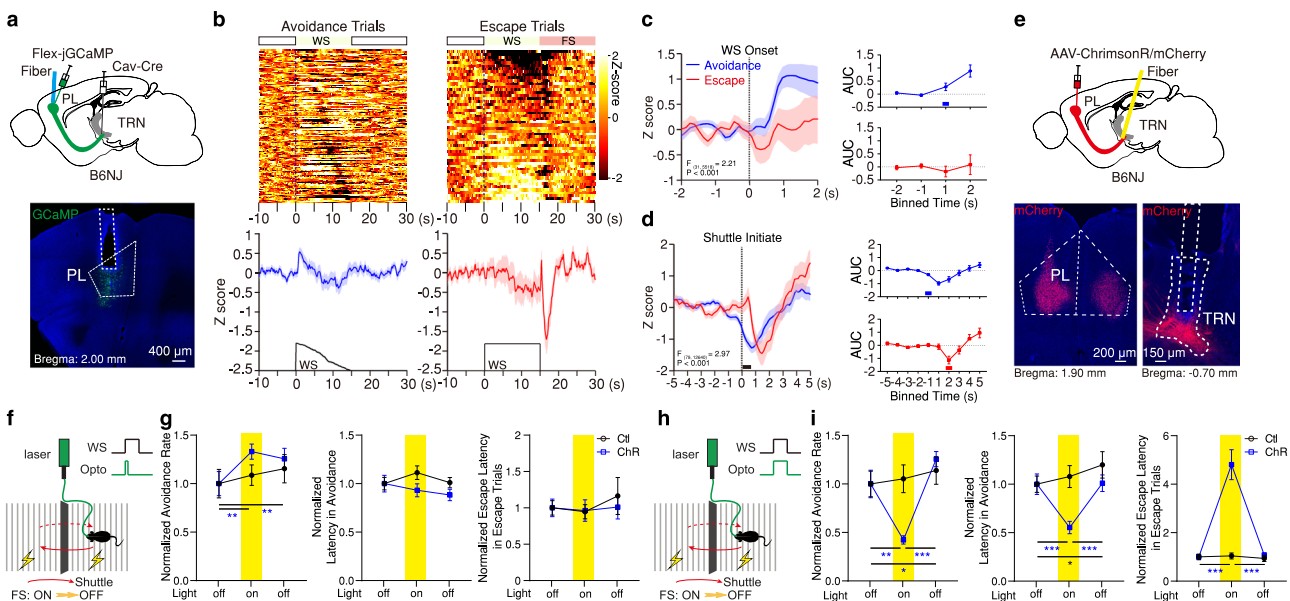

**Fig. 5 | Dynamic activity in PL–avTRN neurons signal the decision to and initiation of active avoidance. a** Top: Schematic of the experimental approach for fiber photometry imaging of avTRN-projecting PL neurons. Bottom: Representative images of GCaMP8s expression and optical fiber placement in the PL region. **b** Top: Heatmaps of calcium responses for avoidance and escape trials. Bottom: Average calcium signal and WS duration, $n = 6$ mice. **c, d** Left: Averaged calcium responses for avoidance (blue) and escape (red) trials during WS onset (c) and shuttle initiate events (d). Mixed-effects model (REML) interactions reported in graphs. WS onset: Avoidance, $n = 128$ events; Escape, $n = 52$ events. Shuttle initiate: Avoidance, $n = 112$ events; Escape, $n = 50$ events. Right: Calcium signal AUC quantification in 1 s bins. WS onset REML interaction, $F_{(3,534)} = 2.60$, $P = 0.051$; Shuttle Initiate REML interaction $F_{(9,1440)} = 3.20$, $P = 0.0008$. **e** Top: Viral strategy schematic for optogenetic stimulation of PL–avTRN terminals. Bottom: Representative images of PL mCherry expression and fiber placement around avTRN. **f** 2AA Task schematic. **g** Group behavioral data across test sessions normalized to the first light-off session for

avoidance rates (left), latencies to avoid (middle), and latencies to escape (right). Two-way ANOVA interactions provided ($n = 11$ mice per group). Avoidance rate: $F_{(2,40)} = 1.74$, $P = 0.19$. Latency to avoid: $F_{(2,40)} = 2.23$, P = 0.12. Latency to escape: $F_{(2,58)} = 0.31$, $P = 0.73$. **h** 2AA task schematic. **i** Group behavioral data across test sessions normalized to the first light-off session for avoidance rates (left), latencies to avoid (middle), and latencies to escape (right). Two-way ANOVA interactions provided (Ctl: $n = 9$ mice; ChR: $n = 8$ mice). Avoidance rate: $F_{(2,30)} = 18.88$, $P < 0.0001$. Latency to avoid: $F_{(2,29)} = 7.66$, $P = 0.0007$. Latency to escape: $F_{(2,28)} = 35.59$, $P < 0.0001$. For all quantifications, multiple comparisons were conducted corrected by a two-stage linear step-up procedure, black lines along the x-axis indicate significant changes reported between groups, and red or blue lines denote the first significant change from the previous bin for within trial type comparisons. Asterisks indicate where ANOVA multiple comparisons found significance. Data are shown as mean ± s.e.m. Source data are provided as a Source Data file.

responses (Fig. 4), our findings suggest that L5 PL neurons might be responsible for the selective recruitment of the avTRN during avoidance trials. Accordingly, when specifically monitoring the activity of projections from L5 PL neurons to the avTRN, we observed similar WS-related dynamics (Supplementary Fig. 11). Collectively, our findings suggest that recruitment of PL→avTRN neurons promotes avoidance decisions. In agreement with this view, brief optogenetic stimulation of the PL→avTRN pathway (2 s) at the WS onset improved active avoidance (Fig. 5e–g; Supplementary Fig. 10e, f; Supplementary Fig. 12a, 8d–f), a result which is consistent with the notion that transient activation in this pathway at WS onset is a signature of avoidance (Fig. 5c). In addition to differences in threat representation between the PL→PVT and PL→avTRN (particularly L5-derived projections) pathways (WS response), inhibitory responses in PL→avTRN neurons emerged -1 s prior to the onset of avoidance responses but not the initiation of escapes, which unlike avoidance responses are not planned actions but instead are oriented reactions driven by the noxious FS[64-67] (Fig. 5d). Of note, sustained optogenetic manipulations of PL→avTRN projections for the duration of the WS attenuated the rate of avoidance but did not significantly affect freezing in responses to the WS (Fig. 5h, i; Supplementary Fig. 10g–k; Supplementary Fig. 12b, c, 8d–f). Paradoxically, while diminishing avoidance, optogenetic stimulation of PL→avTRN afferents for the duration of the WS also decreased the average latency to shuttle for avoidance trials (Fig. 5i). Together, our findings suggest that the PL→avTRN pathway plays a critical role in guiding avoidance decisions when threat cues emerge.

To determine whether the avTRN is likely to relay PL-derived information to the dorsal midline thalamus to influence behavioral

decisions, we first used fiber photometry to monitor the activity of avTRN→PVT terminals while mice performed the 2AA task (Fig. 6a, b). While excitatory responses at WS onset were not statistically different between avoidance and escape trials, significant changes in GCaMP fluorescence at WS onset were only evident for avoidance trials (Fig. 6c). Furthermore, inhibitory responses were associated with shuttle initiation for avoidances only, and these inhibitory responses emerged ahead of the onset of behavior, suggesting that as with the PL→avTRN pathway these inhibitory responses are a feature of the decision to engage in an avoidance action (Fig. 6d; Supplementary Fig. 13a–f). Altogether, our findings suggest that similar to L5 PL→avTRN neurons, and unlike direct PL→PVT projections, avTRN→PVT terminals might be engaged in an outcome-dependent manner and support the idea that the avTRN→PVT pathway underlies the selection and execution of avoidance responses. In agreement with this notion, like PL→avTRN projections, brief optogenetic stimulation of the avTRN→PVT pathway at WS onset for 2 s facilitated avoidance, whereas sustained stimulation throughout the entirety of the WS impaired avoidance without impacting freezing (Fig. 6e–i; Supplementary Fig. 13g–k; Supplementary Fig. 14, 8g–i). In addition, unilateral optogenetic inactivation of avTRN terminals (to prevent effects on behavior resulting from bilateral inhibition to become a confounding factor, Supplementary Fig. 15) significantly attenuated task-related dynamics in PVT→NAc neurons, including WS-onset and shuttle-related calcium responses (Fig. 7; Supplementary Fig. 15). Moreover, unilateral inactivation of L5 PL→avTRN projections in 2AA trained mice decreased PVT neuron engagement during the expression of active avoidance behavior (Supplementary Fig. 16). Lastly, preventing

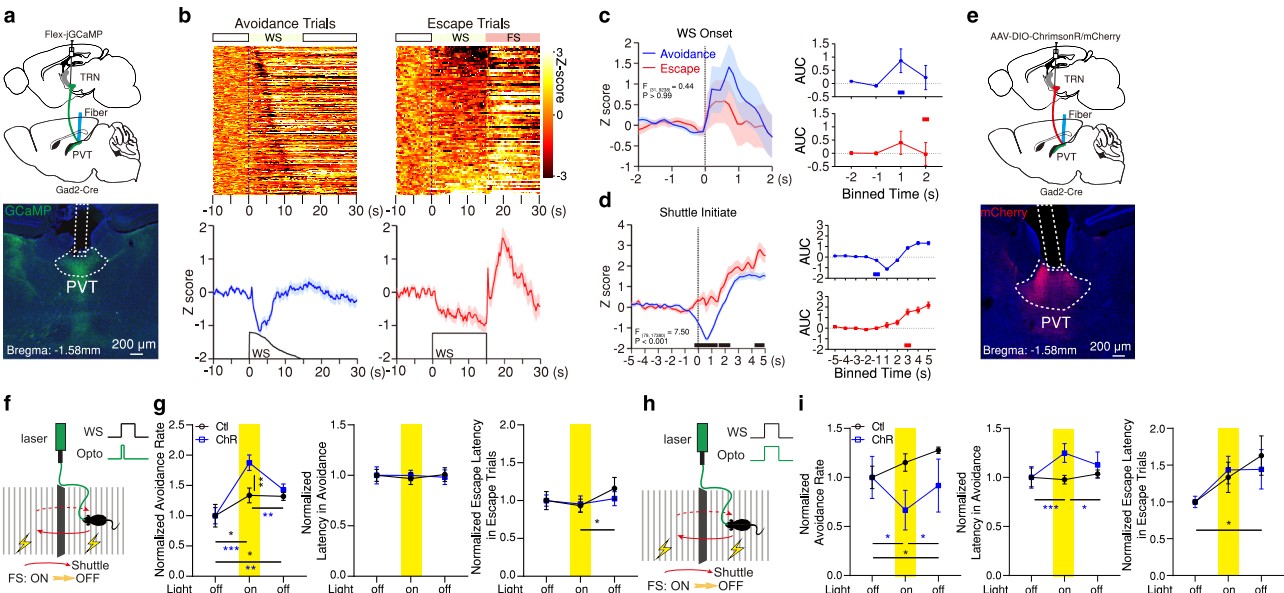

**Fig. 6 | The avTRN–PVT circuit controls active avoidance behavior. a** Top: Schematic of the experimental approach for fiber photometry imaging of the axon terminals of GAD2⁺ TRN cells in the pPVT. Bottom: Representative images of GCaMP7s expression and fiber placement around pPVT. **b** Top: Heatmaps of calcium responses for avoidance and escape trials. Bottom: Average calcium signal and WS duration, $n = 5$ mice. **c**, **d** Left: Average calcium responses for all avoidance (blue) and escape (red) trials during WS onset (**c**) and shuttle initiate events (**d**). Mixed-effects model (REML) interaction reported in graphs. WS onset: Avoidance, $n = 214$ Events; Escape, $n = 86$ events. Shuttle initiate: Avoidance, $n = 158$ events; Escape, $n = 64$ events. Right: Calcium signal AUC quantification in 1 s bins. WS onset REML interaction, $F_{(3,894)} = 0.28$, $P = 0.84$; Shuttle Initiate REML interaction $F_{(9,1980)} = 9.19$, $P < 0.0001$. **e** Top: Viral strategy schematic for optogenetic stimulation of avTRN–PVT circuit. Bottom: Representative image of mCherry-opsin expression and fiber placement around pPVT. **f** 2AA task schematic. **g** Group behavioral data across test sessions normalized to first light-off session for avoidance rates (left), latencies to avoid (middle), and latencies to escape (right). Two-way ANOVA interactions provided, (Ctl, $n = 7$ mice; ChR, $n = 9$ mice). Avoidance rate: $F_{(2, 28)} = 4.73$, $P = 0.017$, Latency to avoid: $F_{(2, 28)} = 0.13$, $P = 0.88$, Latency to escape: $F_{(2, 42)} = 0.94$, $P = 0.4$, **h** 2AA task schematic. **i** Group behavioral data across test sessions normalized to first light-off session for avoidance rates (left), latencies to avoid (middle), and latencies to escape (right). Two-way ANOVA interactions provided (Ctl, $n = 6$ mice; ChR, $n = 7$ mice). Avoidance rate: $F_{(2, 22)} = 4.09$, $P = 0.031$. Latency to avoid: $F_{(2, 21)} = 5.96$, $P = 0.0089$. Latency to escape: $F_{(2, 22)} = 0.39$, $P = 0.68$. For all quantifications, multiple comparisons were conducted corrected by two-stage linear step-up procedure, black lines along x axis indicate significant changes reported between groups, and red or blue lines denote the first significant change from the previous bin for within trial type comparisons. Asterisks indicate where ANOVA multiple comparisons found significance. Data are shown as mean ± s.e.m. Source data are provided as a Source Data file.

avTRN→PVT pathway recruitment during the WS reduced avoidance behavior without impacting freezing (Supplementary Fig. 17). Collectively, our findings support the notion that the avTRN relays PL-derived information to the PVT to shape avoidance decisions.

### Brief engagement of TRN drives PVT activity

While the precise mechanisms by which TRN-generated signals are translated within the PVT require further investigation, the observations described thus far support a model wherein PL-mediated recruitment of the avTRN at WS onset and subsequent inactivation of their thalamic projections may precipitate avoidance-related engagement of the PVT via disinhibition. In partial support of this model, using a recently developed genetically encoded sensor[68] (Supplementary Fig. 18) alongside fiber photometry we monitored changes in GABA release in the PVT during active avoidance. Our findings revealed that shuttle initiation is associated with decreased GABAergic inhibition of PVT→NAc cells (Supplementary Fig. 19). We also observed changes in GABA release associated with other task-related events, including the onset of the WS (Supplementary Fig. 19). Given the complexity and diversity of inhibitory inputs to the PVT[69], it is possible that these changes in GABA concentration reflect modulation of inputs other than the TRN. Altogether, these data suggest that TRN-mediated disinhibition might engage PVT→NAc neurons via post-inhibitory rebound excitation. Of note, PVT neurons are known to display ionic conductances that make them amenable to engaging in post-inhibitory rebound[70]. Importantly, and in agreement with the avTRN's capacity to elicit post-inhibitory rebound responses in the PVT, we had three key findings. First, optogenetic silencing of

avTRN→PVT terminals during 2AA training elicited transient excitation in PVT→NAc neurons at light onset (Fig. 8a–i). Second, optogenetic stimulation of avTRN→PVT terminals in vivo resulted in robust suppression and subsequent rebound activity in PVT→NAc neurons at light offset (Fig. 8j–m). Lastly, pharmacological blockade of GABA receptors in brain slices prevented the avTRN-evoked rebound activity in PVT neurons, supporting the notion that rebounds are triggered by GABAergic mechanisms (Fig. 8n–p). Taken together these experiments suggest that biphasic responses that emerge in PL when threats are signaled (transient increase at WS onset and decrease prior to avoidance initiation) subserve a process by which cortically driven engagement and subsequent disengagement of avTRN neurons triggers instrumental defensive actions via the PVT.

### Discussion

In this study, we investigated the anatomical and functional organization of prefronto-thalamic circuits, focusing on PL projections to the PVT. Our findings yielded several key and unexpected observations. Specifically, we discovered that while PL sends direct projections to the PVT that lead to modest postsynaptic currents in striatal-projecting PVT neurons, PL potently inhibits PVT neurons via a strong di-synaptic pathway that implicates a subpopulation of PV-negative GABAergic cells of the avTRN. In vivo studies further revealed how direct and indirect projections from PL shape goal-oriented defensive actions via the PVT. Thus, whereas direct PL projections modulate the spontaneous activity of PVT neurons and promotes defensive states, indirect PL projections through the avTRN act as a major driver of task-related dynamics in PVT neurons and promote avoidance decisions.

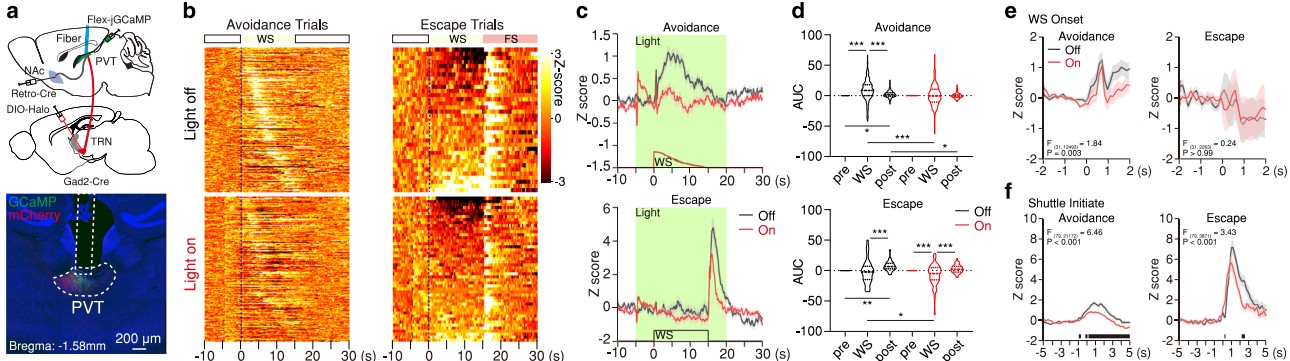

**Fig. 7 | The avTRN shapes avoidance-related dynamics in PVT–NAc neurons.**
**a** Top: Schematic of the experimental approach used for fiber photometry imaging of NAc-projecting PVT cells and optogenetic inhibition of the avTRN–PVT pathway. Bottom: Representative image of GCaMP8s and Halo-mCherry expression, and fiber placement around pPVT. **b** Heatmaps of calcium responses for avoidance and escape trials in light-off and light-on sessions, $n = 3$ mice. **c** Average calcium signal and WS duration in (**b**). **d** Calcium signal AUC quantification for avoidance and escape trials, Mixed-effects model (REML) interactions reported in graphs. Avoidance: Off, $n = 205$ Trials; On, $n = 192$ Trials. Escape: Off, $n = 35$ Trials; On, $n = 48$ Trials. **e–f** Averaged calcium responses during WS onset (**e**) and shuttle initiation (**f**) events for all avoidance and escape trials, $n = 4$ mice. Mixed-effects model (REML) interactions reported in graphs. WS onset: Avoidance, Off $n = 205$ Events, On $n = 192$ Events. Escape, Off $n = 35$ Events, On $n = 48$ Events. Shuttle initiate: Avoidance, Off $n = 139$ Events, On $n = 131$ Events; Escape, Off $n = 24$ Events, On $n = 27$ Events. For all quantifications, multiple comparisons were conducted corrected by two-stage linear step-up procedure, black lines along x axis indicate significant changes reported between groups. Asterisks indicate where ANOVA multiple comparisons found significance. Data are shown as mean ± s.e.m. Source data are provided as a Source Data file.

Consistent with this notion, we found that transient recruitment of the avTRN, resulted in strong post-inhibitory rebound excitation in PVT neurons both in vivo and ex vivo. In summary, our study uncovered fundamental features of the anatomical and functional organization of PL → PVT circuits underlying active avoidance behavior (Supplementary Fig. 20). These findings have important implications for understanding corticothalamic circuitry beyond sensory systems. Indeed, while earlier research predominantly focused on sensory functions, our study highlights the adaptability and complexity of these corticothalamic circuits in the context of higher-order cognitive processes, such as decision-making and emotional regulation. Below, we expand on our main observations and their implications.

### PL exerts driver-like influence over the PVT via the TRN

Our results suggest that the avTRN acts as a major conduit via which the mPFC exerts top-down control of the PVT, thus challenging both conventional and recent views on corticothalamic network organization[39,71]. Indeed, while direct projections from PL to the PVT were relatively modest, mostly L6-derived, and displayed modulator-like control over neuronal activity in the midline thalamus, indirect projections through the avTRN were robust, contained a significant L5 input that encoded warning signal information in an outcome-dependent manner, and significantly contributed to task-related dynamics in PVT neurons. Collectively, these observations support the notion that the avTRN plays a central role in mediating top-down control of the PVT, potentially serving as a hub for integrating information from higher-order cortical regions. Thus, insight generated in the present study expands our understanding of the mechanisms by which the prefrontal cortex influences thalamic processing and highlights the importance of inhibitory mechanisms in shaping thalamic activity[9]. Of note, midline and intralaminar nuclei are known to display highly specialized inhibitory inputs arising from extra-thalamic sources which further highlights the complexity of inhibitory regulation of thalamic circuitry[72]. Overall, our results are consistent with recent findings that unlike traditional model of corticothalamic organization[73,74], L5 neurons innervate higher-order sectors of the avTRN[38,39,75]. Furthermore, our findings that direct (mostly L6-derived) and indirect (including both L5 and L6) projections of PL differentially modulate spontaneous activity and task-related responses in the PVT are consistent with recent reports on the functional features of parallel cortical streams to visual thalamic areas[76]. Importantly, we expand on this framework by proposing an updated model where L5 corticothalamic inputs can influence thalamic nuclei that are largely devoid of L5 innervation by driver-like signals channeled through the TRN.

Our study's intriguing finding that TRN projections, although primarily inhibitory[77], mediate task-related increases in neuronal activity in avoidance-promoting PVT neurons appears paradoxical when compared to studies in sensory systems where optogenetic stimulation of cortex leads to profound inhibition of thalamic relays[36,78]. However, we provide several lines of evidence supporting our conclusion. First, transient optogenetically-evoked activation of avTRN afferents to the PVT elicited strong post-inhibitory rebounds in vivo, which were confirmed to depend on local GABAergic signaling in acute brain slices. These effects are likely to be entirely mediated by monosynaptic input from the avTRN because thalamic nuclei in rodents (including the PVT) are largely devoid of GABAergic interneurons[79]. Second, inhibition of either the PL→avTRN pathway or avTRN→PVT projections in vivo diminished, rather than enhanced, the activation of PVT neurons during active avoidance. Lastly, the idea that post-inhibitory rebounds might be a major mechanism by which the avTRN exerts driver-like influence over the dorsal midline thalamus agrees with the known membrane properties of thalamic relays including PVT neurons[70,80]. Despite the evidence provided, future studies should focus on elucidating the cellular and molecular mechanisms underlying avTRN inputs' modulation of neuronal activity in the PVT, including the rebound actions reported here.

In addition to the above-described features, the in vivo dynamics of the TRN-mediated PL input to the PVT suggested that while this pathway was selectively recruited by the WS onset during avoidance trials (including L5-mediated projections to the avTRN), the initiation of avoidance responses was reliably preceded by inhibitory signals within this pathway. This observation suggests that timed suppression of avTRN-derived input following activation by the WS is conducive to increased activity in PVT neurons and the subsequent initiation of avoidance responses. Accordingly, disrupting either aspect of this activity hindered avoidance. We hypothesize that this TRN-mediated disinhibitory mechanisms likely belongs to a broader architecture via which cortical inputs exerts top-down control over the dorsal midline thalamus. A compatible proposal can be found in a recently published study where TRN-mediated disinhibition was speculated to engage the

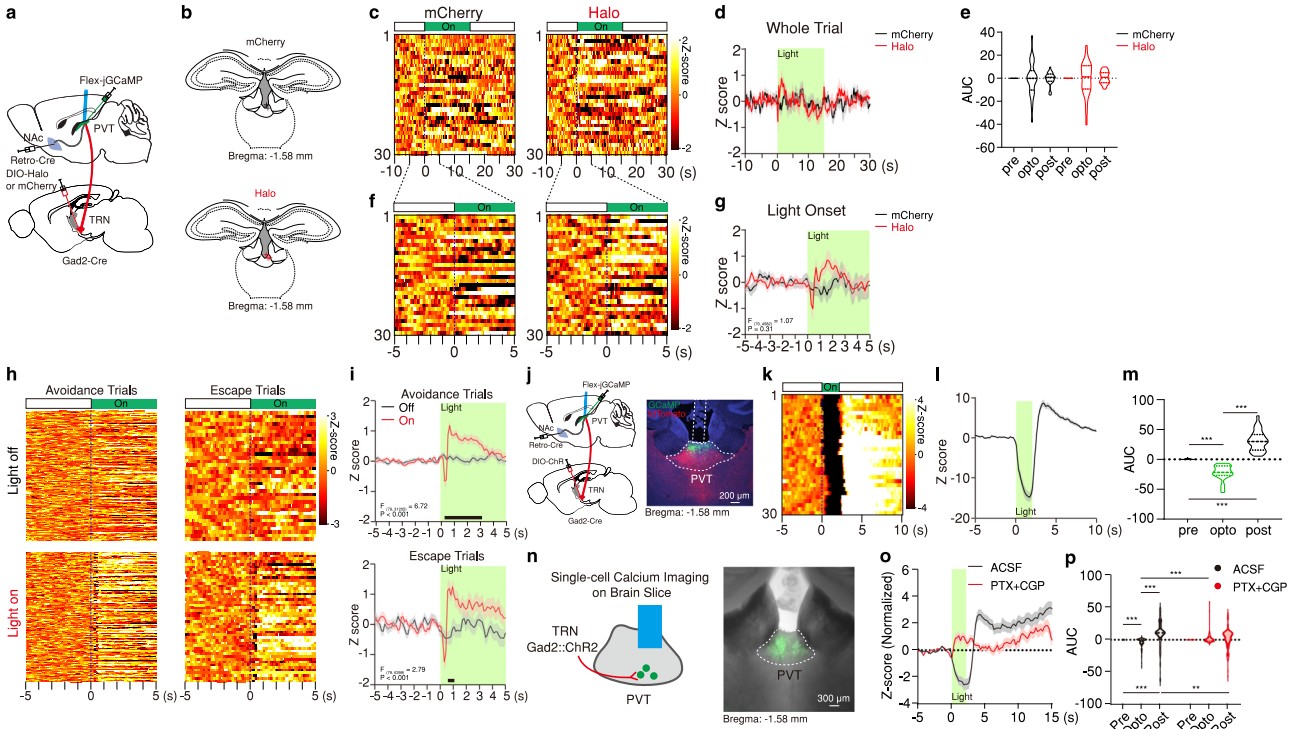

**Fig. 8 | Optogenetic manipulations of input from avTRN drive rebound responses in NAc-projecting PVT neurons. a, b** Schematic of the experimental approach for fiber photometry imaging of NAc-projecting PVT cells while opto-genetically inhibiting avTRN-PVT pathway, and fiber placement around pPVT ($n = 3$ mice per group). **c, d** Heatmaps of calcium responses for mCherry and Halo subjects (left two) and average calcium signal (right) before 2AA. **e** Calcium signal AUC quantification for mCherry and Halo subjects, two-way ANOVA interaction provided ($n = 30$ Trials from 3 mice for each group); $F_{(2, 116)} = 0.08$, $P = 0.92$. **f, g** Heatmaps (**f**) and averages (**g**) of calcium responses for mCherry and Halo subjects from pre 5-s to post 5-s of light onset from (**c**). Mixed-effects model (REML) interaction reported in graph. **h, i** Adapted from Fig. 7b, during 2AA. Heatmaps (**h**) and averages (**i**) of calcium responses for Light-off and Light-on sessions from 10-s prior to WS onset. Mixed-effects model (REML) interaction reported in graphs, Avoidance: (Off, $n = 205$ Trials; On, $n = 192$ Trials); Escape: $P < 0.001$ (Off, $n = 35$ Trials; On, $n = 48$ Trials). **j** Top: Schematic of the experimental approach for fiber photometry imaging of NAc-projecting PVT cells and optogenetically activating

avTRN-PVT pathway. Bottom: Representative images of GCaMP7s and ChrimsonR-tdTomato expression, and fiber placement around pPVT. **k** Heatmap of calcium responses during optical stimulation. **l** Average calcium signal. **m** Calcium signal AUC quantification for each stage. One-way ANOVA interaction; $F_{(2, 87)} = 144.2$, $P < 0.0001$ ($n = 30$ Trials from 3 mice). **n** Left: Schematic of the experimental approach for ex vivo single-cell slice imaging of GCaMP during TRN optogenetic activation. Right: Example 4x image of slice with GCaMP8f expression. **o** Averaged z-score calcium signal before and after bath application of PTX (50 μM) and CGP55845 (5 μM) during optogenetic stimulation of TRN normalized to the slope of baseline. **p** Calcium signal AUC quantification before and after bath application of GABA antagonists. Two-way repeated measures ANOVA interaction, $F_{(2, 266)} = 31.74$, $P < 0.0001$ ($n = 134$ cells from 5 mice). For all quantifications, multiple comparisons were conducted corrected by two-stage linear step-up procedure, black lines along x-axis indicate significant changes reported between groups. Asterisks indicate where ANOVA multiple comparisons found significance. Data are shown as mean ± s.e.m. Source data are provided as a Source Data file.

PVT during conditioned freezing[29]. Additionally, inhibitory responses in PL neurons have been associated with the onset of defensive behaviors[15,29,81]. It is worth mentioning that the source of this inhibition to PL and/or the TRN thought to underlie the initiation of defensive actions via the midline thalamus remains unknown and should be the focus of future studies. One possibility is that local interneurons within PL contribute to this process. Indeed, molecularly defined sub-populations of GABAergic interneurons within PL have been shown to shape conditioned freezing behavior[82,83] and may play a broader role in shaping defensive behaviors.

## PL is important for the construction of fear states and the selection of adaptive responses

A substantial body of literature underscores the critical role for PL in processing threats and guiding defensive behaviors[21,84]. Previous studies have demonstrated that PL is essential for fear memory retrieval and contributes to both conditioned freezing and active avoidance behavior in rodents[14,82–90]. However, recent reports indicating the absence of outcome-dependent threat representations in PL neurons appear contradictory to this notion[86]. Our study addresses this discrepancy by revealing the presence of such representations in PL neurons projecting to the avTRN, suggesting a potentially unique

characteristic of this specific cortical pathway most likely due to the inclusion of an L5-derived input. This underscores the functional diversity within prefrontal circuits involved in emotional processing. Moreover, our findings suggest that the distinct modulatory and driver-like effects of direct and indirect projections may be attributed to the differential contributions of L6 and L5-derived information from PL, respectively. While direct input primarily from L6 appears to influence the spontaneous activity of PVT neurons and may contribute to promoting defensive states, indirect projections through the TRN seem to predominantly facilitate the subjects' capacity to plan and execute avoidance responses. Of note, while our study sheds light on these complex cortical-thalamic circuits, further investigation is warranted to dissect the contribution of specific laminar within both direct and indirect projections to the PVT. Future studies employing transgenic mouse lines that enable genetic access to L5 and L6 neurons of the cortex will be instrumental in elucidating the distinct roles of these cortical layers in shaping avoidance-related neural activity and behavior mediated by the PVT.

## Limitations of the present study

While our study provides valuable insights into the functional organization of prefronto-thalamic circuits, it is important to acknowledge

certain limitations. Firstly, we employed fiber photometry to measure bulk changes in activity. While this technique offers the advantage of recording the activity of genetically defined neuron subpopulations and tracking axon terminal activity, it suffers from poor temporal resolution[91]. Additionally, the use of fiber photometry captures the summed activity of many neurons, potentially masking the heterogeneous activity dynamics of individual neurons within the recorded population. These aspects of the technique may account for the variability in the temporal structure of calcium signals observed across the various nodes investigated in our study or small but relevant differences in transient activity seen at WS onset for avTRN terminals in PVT. In vivo electrophysiology would provide a more suitable approach to explore the temporal structure of activity in direct and indirect cortical streams to the PVT. Furthermore, as noted earlier, our study indicates that the actions of direct and indirect projections may be attributed to specific lamina of the PL. However, future studies employing precise manipulations of specific laminae of the PL will be necessary to dissect the functional contributions of each layer with greater precision.

It is also worth noting that while our identification of a subset of *Gad2*+ avTRN neurons that do not express PV finds support in recent literature[44,46,47], one possibility is that cells recorded in our study belong to neighboring areas such as the lateral hypothalamic area (LH) and the zona incerta (ZI), both of which project to the PVT[26,50–53]. However, this scenario is unlikely given that we specifically targeted the portion of the TRN located in the most anterior parts of the thalamus and directly above the stria medullaris (sm), a region consistently considered part of the TRN in mouse atlases. This targeted region is distinct from the LH, which is located below the sm, and the ZI, which is posterior to the avTRN and more ventromedially located.

### Active avoidance: a defensive response to low-imminence threats

In nature, defensive behaviors exhibit significant variation based on the spatiotemporal proximity of threats[92]. When threats are imminent, animals swiftly deploy defensive behaviors from their innate repertoire. These high-threat imminence states trigger responses like freezing behavior and escape actions in rodents. Freezing behavior occurs when the animal perceives an immediate threat in its surroundings, while escape actions are elicited when the threat is in direct contact with the animal, such as during a predatory attack[64–67]. In contrast, when threats are perceived but not yet imminent, animals engage in pre-emptive defensive strategies. This can involve non-defensive survival actions when safety is presumed or avoidance actions when threats are anticipated but not yet present[93–96].

Neuroscientific research into the neural mechanisms underlying these different defensive responses in rodents has yielded a model that predominantly aligns with the temporal constraints and demands associated with their deployment. Defensive reactions like freezing and escape are primarily initiated by subcortical areas, such as the superior colliculus and the periaqueductal gray[97,98]. In contrast, pre-emptive strategies like active avoidance and risk assessment are believed to involve more cognitive processing, implicating the cortex[99,100]. Notably, even when defensive behaviors originate from subcortical regions under high threat imminence, cognitive processes play a significant role in shaping the intricate details of these responses[101,102].

The PVT has been implicated in guiding defensive behaviors under both high and low-threat imminence. For instance, the PVT is essential for conditioned freezing behavior and plays a key role in driving avoidance responses under situations of high and low conflict[28,40,41,103,104]. In our study, we have demonstrated that top-down modulation from the mPFC supports active avoidance behavior through the PVT. Notably, this prefrontal connectivity with the PVT also influences the expression of conditioned freezing[28]. While it may

seem unexpected that the mPFC, traditionally associated with higher functions, is required for freezing behavior, recent findings suggest that the mPFC is primarily involved in the maintenance rather than the initiation of freezing bouts[105]. This aligns with the emerging perspective that freezing is not merely a passive, reactive response but is closely tied to cognitive processes.

Furthermore, it is worth noting that our study expands the role of limbic TRN to include active avoidance behavior, adding to prior research that has documented a role for TRN, particularly PV+ neurons, in shaping defensive reactions such as freezing and escape behavior via projections to midline thalamic regions[35,106]. Collectively, these studies highlight the importance of this brain axis as a critical hub shaping the activity of thalamic relays and defensive responses across multiple spatiotemporal scales of threat imminence. This underscores the versatility and adaptability of thalamic circuitry involving TRN in orchestrating defensive behaviors, ranging from immediate reactive responses to pre-emptive avoidance strategies, thus ensuring the organism's survival in diverse threat environments.

In conclusion, our collective results have provided valuable insights into the mechanisms through which top-down modulation of the dorsal midline thalamus promotes defensive actions. Given that mPFC-derived cortical input shapes emotional and motivated behaviors via the PVT, the mechanisms uncovered by our research hold significant potential for understanding and potentially influencing these critical processes.

## Methods

### Mice

All procedures were performed in accordance with the *Guide for the Care and Use of Laboratory Animals* and were approved by the National Institute of Mental Health (NIMH) Animal Care and Use Committee. Mice used in this study were group housed under a 12-h light/dark cycle (6:00–18:00 light), at temperatures of 70–74 °F and 40–65% humidity, with food and water available *ad libitum*. After surgery, mice were singly housed. C57BL/6NJ (005304), *Gad2*-Cre (019022), and *Pv*-Cre (017320) were obtained from The Jackson Laboratory. *Rbp4*-Cre (founder line KL100), *Drd2*-Cre (ER44), and *Syt6*-Cre (KI148, 109, or 130) were obtained from GENSAT/MMRRC. Both male and female mice were used across all experiments and combined in analyses as no statistically significant effect of sex were found in the variables tested in pilot studies. Animals were randomly allocated to the different experimental conditions reported in this study. For all behavioral manipulations and fiber photometry experiments mice were handled for at least 3 consecutive days prior to testing.

### Viral vectors

AAV2-CaMKII-eNpHR3.0-mCherry (Deisseroth), AAV2-Ef1α-DIO-hChR2 (H134R)-EYFP (Deisseroth), AAV5-CaMKII-ChR2(H134R)-EYFP (Deisseroth), AAV2-CaMKII-EYFP (Deisseroth), AAV2-Ef1α-DIO-eNpHR3.0-mCherry (Deisseroth), and AAV2-syn-FLEX-ChrimsonR-tdTomato (Boyden) were produced by the Vector Core of the University of North Carolina. AAV9-hSyn-Flex-GCaMP8s-WPRE (plasmid no. 162377), AAV9-syn-ChrimsonR -tdTomato (plasmid no. 59171), AAV2-CaMKII-mCherry (plasmid no. 114469), AAV2(retro)-CAG-iCre (plasmid no. 81070), AAV2-hSyn-DIO-hM4Di-mCherry (plasmid no. 44362), AAV2-hSyn-DIO-mCherry (plasmid no. 50459), AAV9-syn-FLEX-jGCaMP7s-WPRE (plasmid no. 104491), and AAV1-hSyn-FLEX-iGABASnFR (plasmid no. 112163) were purchased from Addgene. CAV-Cre was produced by the Institute of Molecular Genetics of Montpellier (Montpellier, France). AAV9-EF1a-FLEX-TVA-mCherry (Addgene, plasmid no. 38044) and AAV9-CAG-FLEX-RG (Addgene, plasmid no. 38043) were produced by Vigene Biosciences. EnvA-SAD-ΔG-eGFP (Addgene, plasmid no. 32635) was produced by the Viral Vector Core of the Salk Institute for Biological Studies. All viral vectors were stored in aliquots at −80 °C until use.

## Stereotaxic surgery

All stereotaxic surgeries were conducted as described in our animal study protocol using well-documented procedures and stereotaxic coordinates[103,106]. Mice were first anesthetized with a ketamine (100 mg/kg) plus xylazine (10 mg/kg) solution and placed in a stereotaxic device (AngleTwo, Leica Biosystems). The following stereotaxic coordinates were targeted for viral injections and/or optical fiber implantation: PL, −1.90 mm from bregma, ±0.55 mm lateral from midline and −2.30 mm vertical from cortical surface; avTRN, −0.70 mm from bregma, ±1.00 mm lateral from midline and −4.20 mm vertical from cortical surface; pPVT, −1.60 mm from bregma, 0.06 mm lateral from midline and −3.30 mm vertical from cortical surface, 6.12° angle for both fiber photometry and optogenetics; NAc, 1.70 mm from bregma, ±0.60 mm lateral from midline and −4.80 mm vertical from cortical surface.

For fiber photometry and optogenetic experiments, an optical fiber (400 μm for photometry, Doric Lenses; 200 μm for optogenetics, ThorLabs) was implanted 200–300 μm above the target of interest and immediately following viral injections, and subsequently cemented using Metabond Cement System (Parkell) and Jet Brand dental acrylic (Lang Dental Manufacturing).

For retrograde tracing of TRN-projecting and PVT-projecting PL cells CTB-488 and CTB-555 (1.0% in PBS, Thermo Fisher Scientific) were injected into the TRN (0.5 μL) and PVT (0.6 μL), respectively, and allowed 4 d for retrograde transport. For retrograde tracing of PVT-projecting TRN cells unconjugated CTB (List Labs Product No. 104) was injected into the PVT (0.6 μL). For retrograde labeling of PVT-projecting TRN cells for RNAscope, retrobeads (0.6 μL, LumaFluor, Inc.) were injected into the PVT and allowed 7 d for retrograde transport.

After all surgical procedures, animals were returned to their home cages and placed on a heating pad for 24 h for post-surgical recovery and monitoring. Animals received subcutaneous injections with Metacam (meloxicam, 5 mg/kg) for analgesia and anti-inflammatory purposes. Mice without correct targeting of optical fibers, tracers, or vectors were excluded from this study.

## Fiber photometry

Fiber photometry was performed in accordance with previously described methodological procedures[69] and are detailed below. Mice were allowed to habituate to the fiber patch cord in their home cage for approximately 5 min before each behavior test. GCaMP fluorescence and isosbestic autofluorescence signals were excited by the fiber photometry system (Doric Lenses) using two sinusoidally modulated LEDs (473 nm at 211 Hz and 405 nm at 531 Hz) controlled by a standalone driver (DC4100, ThorLabs). Both LEDs were combined via a commercial Mini Cube fiber photometry apparatus (Doric Lenses) into a fiber patch cord (400-μm core, 0.48 NA) connected to the brain implant in each mouse. The light intensity at the interface between the fiber tip and the animal was adjusted from 10 μW to 20 μW (but was constant throughout each test session for each mouse). An RZ5P fiber photometry acquisition system with Synapse software (Tucker-Davis Technologies) collected and saved real-time demodulated emission signals and behavior-relevant TTL inputs. For each trial, GCaMP signals ($F_{473 nm}$) were compared with autofluorescence signals ($F_{405 nm}$) to control for movement and bleaching artifacts. Signal data were detrended by first applying a least-squares linear fit to produce $F_{fitted 405 nm}$, and dF/F was calculated as ($F_{473 nm} − F_{fitted 405 nm}$)/$F_{fitted 405 nm}$. All GCaMP signal data are presented as the z-score of the dF/F from baseline (pre-WS) segments. All heat maps for avoidance trials are organized by latency from shortest to longest, and all heat maps for escape are organized by overall response changes from baseline during WS from reduced to increased.

## Two-way active avoidance (2AA)

Mice were trained on the 2AA task replicating previous methodological procedures[40] and are described herein. The behavioral apparatus consisted of a custom-built shuttle box (18 cm × 36 cm × 30 cm) that contained two identical chambers separated by a hurdle (17.5 cm × 6 cm). The hurdle projected 3 cm above the floor and allowed mice easy access to both chambers. The floor consisted of electrifiable metal rods (H10-11M-TC-SF, Coulbourn Instruments) and was connected to a shock generator (H13-15, Coulbourn Instruments). Before each subject was trained/tested, the shuttle box was wiped clean with 70% ethanol. The mouse's behavior was captured with a USB camera during each session. A speaker located on the top of the shuttle box (50 cm high) was used to deliver the WS. Subjects' movement and TTLs of WS, US and optogenetic stimulation were recorded by ANY-maze version 5 (Stoelting).

After a 5-min habituation period, mice were trained with daily sessions of 2AA, each consisting of 30 presentations of the WS (4 kHz, 75 dB, lasting up to 15 s each). Trials in which subjects failed to shuttle to the adjacent chamber before the termination of the WS resulted in the presentation of the US (0.6 mA foot shock lasting up to 15 s each) until subjects escaped to the opposite chamber (escape trials). No subject failed to escape the US. For trials in which subjects shuttled to the opposite chamber during the WS, the WS was abruptly terminated, and the US was also prevented (avoidance trials). The inter-trial interval (ITI) was 30 s. Avoidance rate was calculated as the percentage of the number of avoidance trials over the total number of trials. For optogenetic and fiber photometry experiments fiber patch cords were attached every session of training. For all experiments, subjects that did not reach 30% avoidance rates by Day 5 were excluded from data analysis. All experiments were conducted in awake, freely moving mice in test chambers during the light cycle.

**Fiber photometry during 2AA.** In Fig. 4a–d, Fig. 5a–d, Supplementary Fig. 4 and Supplementary Fig. 10a–d, mice were subjected to five 2AA sessions (one session per day), and the GCaMP signal was collected on Day 5 as described above. In Fig. 6a–d, Supplementary Fig. 11, Supplementary Fig. 13a–f, and Supplementary Fig. 19, mice were subjected to five 2AA sessions (one session per day), and the GCaMP or iGA-BASnFR signal was collected on Days 4 and 5 as described above.

**Optogenetic manipulations during 2AA.** In Fig. 4e, f, Fig. 5h, i, Fig. 6h, i, Supplementary Fig. 6, Supplementary Fig. 9, Supplementary Fig. 10g–k, Supplementary Fig. 12b, c, Supplementary Fig. 13j, k, Supplementary Fig. 14 and Supplementary Fig. 17f–j, mice were subjected to five 2AA sessions (one session per day), and on Day 4 a yellow light (Ce:YAG + LED Driver, Doric) was presented coinciding with the WS to either inhibit (Halo) or activate (ChrimsonR, ChR), PL fibers in PVT, PL fibers in TRN or TRN fibers in PVT. In Fig. 5f, g, Fig. 6f, g, Supplementary Fig. 10e, f, Supplementary Fig. 11h, i, Supplementary Fig. 12a, and Supplementary Fig. 17a–e, mice were subjected to five 2AA sessions (one session per day), and on Day 4 the yellow light was presented at the onset of the WS but terminated after 2 s. In Supplementary Fig. 7, mice were subjected to five 2AA sessions (one session per day), and on Day 4 the yellow light was presented coinciding with the ITI. For all mice the light intensity at the interface between the fiber tip and the brain implant was -10 mW. For optogenetic excitation (ChR) the light (10 Hz, 20% duty cycle, for both PL-TRN and PL-PVT circuits; 50 Hz, 10% duty cycle, for TRN-PVT circuit only) was presented at the certain duration as each experiment. For optogenetic inhibition (Halo) the light was presented continuously for the certain duration as each experiment.

**Fiber photometry with optogenetic manipulations during 2AA.** In Fig. 4g–l, Fig. 7, Supplementary Fig. 5e, f, and Supplementary Fig. 15, mice were subjected to seven 2AA sessions (one session per day), the GCaMP signal was collected on Days 4–7 and optogenetic inhibition was performed on Days 4 and 6 with constant light presentation that began 5 s prior to WS onset and culminated 5 s after the offset of the WS.

## Data analysis and behavioral tracking for 2AA

Analysis of the 2AA behavioral tracking was done replicating previous methodological procedures[40] and are described herein. We performed post hoc position tracking of the animal's nose and body center from video in the software TopScan (CleverSys). WS and US times from ANY-maze and raw video tracking position values from TopScan were exported, and analysis was performed with custom routines in the R statistical computing environment (R Core Team 2019, R Foundation). Missing positions up to ten successive frames were linearly interpolated with custom routines in R. For imaging sessions, video tracking and ANY-maze TTL pulse timestamps were zero corrected to align behavioral and calcium signal timestamps. Next, calcium signals and/or position frames during US and WS were flagged by matching the relevant timestamps to TTL pulse times from ANY-maze, and the frame-by-frame distance traveled for the nose and body center was calculated for the tracking data. To minimize the effects of noise in the tracking data, we calculated the 40% quantile of the frame-by-frame distance traveled by the animal's nose and body center for each session; in all cases, this yielded a distance value of 0 or 1 mm. This quantile value served as a movement threshold—that is, an inter-frame distance traveled less than or equal to the quantile value was considered non-movement. We then created a binary vector, and frames with coincident immobility of the nose and center body were set to 1. Changepoint analysis[107] (R package version 2.2.2 (https://CRAN.R-project.org/package=changepoint)), with a minimum segment length of 30 video frames, was then applied to this vector. This approach allowed us to statistically determine when transitions to (and from) coincident periods of non-movement of nose and body occurred, which were used as a proxy for freezing behavior. Next, each sustained bout of non-movement was isolated, and we probed whether there was any movement that lasted for >5 consecutive video frames. If such movement did occur, we truncated the bout of immobility at the start of movement. Finally, immobility bouts with a duration ≥1 s were considered freezing.

We isolated freezing events (see freeze detection section above) that occurred during the WS as WS Freezing and those that occurred during ITI as ITI Freezing. For each trial, we calculated the time interval between the moment of animal crossing the hurdle and the WS onset during trials where the animal avoided the footshock, named latency to avoid. WS, ITI Freezing or Latency to Avoid were average within session and then within each group and plotted as mean ±s.e.m.

For fiber photometry, GCaMP data were normalized as dF/F. Next, we used the behavioral flags calculated from the video tracking to create average peri-event time histograms (PETHs) time-locked to the onset of the behavior events of interest, including *WS onset*, highest movement velocity during the WS (*Max Velocity*), escape or avoidance movement onset (*Shuttle Initiate*), escape or avoidance moment (*Shuttle*) and freezing onset during the WS (*WS Freezing*). For clarification, *Max Velocity* is measured during the WS and typically occurs when the mouse is shuttling to the other chamber in avoidance trials, *Shuttle Initiate* refers to the moment in time when the animal begins a movement sequence that results in a shuttle, whereas *Shuttle* refer to the moment in which the subject has crossed the hurdle and either the WS or FS have been terminated as a result. Of note, these three events typically occur relatively close in time since animals usually complete shuttles within approximately one second of initiating a response at maximum velocity during the WS in avoidance trials. All trials in each session were separated into avoidance and escape trials as described above. For each trial type, the z-score from 10 s before to 30 s after WS onset was plotted in heat maps for all trials in test sessions. The mean of all recorded activity for each trial type was plotted below the corresponding heat map. Heat maps are arranged by avoidance latency for avoidance trials and by change in activity from inhibition to activation during the WS for failure trials. We calculated and plotted the AUC of the z-scores before WS onset as a baseline, during the WS, and

post-WS period. We isolated the peri-event calcium signal by a certain time window (2 s for the onset of the WS and 5 s for all other behavioral events) from avoidance and escape trials separately, then we calculated z-scores based on pre-event signal for each trial and area under the curve (*AUC*) of the z-score from 1 s bins throughout each behavior event. Lastly, we plotted the mean of signal transitions and AUC for each event type from each trial type and compared the differences of AUCs of 1) adjacent seconds in same trial type and 2) same second between trial types or groups. All 2AA photometric signals and behavioral performance were analyzed blinded. We compared differences between trial types at each moment by Mixed-effects model (REML) multiple comparisons corrected by two-stage linear step-up procedure of Benjamini, Krieger and Yekutieli (Significant changes were highlighted by black thick lines along x-axis where $P < 0.05$). We also applied comparisons within each trial type between each bin's AUC and the bin immediately preceding it. Blue or red thick lines along x-axis indicate where post hoc tests found the first significant change from the previous bin within trial types ($P < 0.05$, Mixed-effects model (REML) multiple comparisons corrected by two-stage linear step-up procedure of Benjamini, Krieger and Yekutieli; blue line for avoidance trial, red line for escape trial). All codes are available at the following repository: https://github.com/Penzolab/Data-analysis-of-Two-way-active-avoidance-task.git. (https://doi.org/10.5281/zenodo.12707790).

## Fiber photometry with optogenetic and chemogenetic manipulations

In Fig. 3 and Supplementary Fig. 3, mice with mCherry or DREADDs (Gi) unilaterally expressed in the TRN and channelrhodopsin (ChR2) or YFP unilaterally expressed in the PL were injected with either clozapine N-oxide (CNO) or saline (Sal) and subjected to 15 trials of optogenetic stimulation ( ~ 10 mW, 10 Hz, 20% duty cycle, 15 s) with 30 s ITI while simultaneously recording GCaMP signal as described above in NAc-projecting pPVT cells. Both CNO (10 mg/kg; Enzo Life Sciences) and Sal injections were given to all mice 30 min prior to recording GCaMP signal and the injection order was counterbalanced on separate days.

## Monosynaptic rabies tracing of inputs to PVT-projecting TRN cells

For Fig. 1i–k and Supplementary Fig. 1, TRIO rabies tracing was conducted as previously reported and described herein[40,108]. To limit monosynaptic rabies tracing to PVT-projecting neurons of the TRN, CAV-Cre virus was unilaterally injected into the PVT (1.0 µl) of C57BL/6NJ mice. Within the same surgical procedure, a virus mixture of AAV9-EF1a-FLEX-TVA-mCherry and AAV9-CAG-FLEX-RG at a 1:1 ratio was injected into the avTRN (0.6 µl), followed by an injection of the pseudotyped rabies virus EnvA-SAD-ΔG-eGFP (1.2 ul) in the same location of the avTRN 2 weeks later. Mouse brain tissues were collected and subjected to analysis 1 week later. Representative images were scanned with a Zeiss 780 confocal microscope. A similar procedure was used for rabies-assisted tracing of input onto PVT–NAc neurons (Supplementary Fig. 1). For Supplementary Fig. 1 images were aligned to an atlas and automatic cell detection and quantification in a layer specific manner was achieved using NeuroInfo (MBF Bioscience). We generated a connectivity index by normalizing the fraction of retrogradely labeled (GFP+) cells for a given brain region to the number of starter cells[40]. Starter cells count was generated by quantifying the number of mCherry and GFP double positive avTRN (Fig. 1) and pPVT cells (Supplementary Fig. 1).

## Histology and immunofluorescence

Animals were deeply anesthetized with euthanasia solution (Vet One) and transcardially perfused with PBS (pH 7.4, 4 °C), followed by paraformaldehyde (PFA) solution (4% in PBS, 4 °C). After extraction, brains were post-fixed in 4% PFA at 4 °C for a minimum of 2 h and subsequently cryoprotected by transferring to a 30% PBS-buffered

sucrose solution until brains were saturated (for over 24 h). Coronal brain sections (50 μm) were cut using a freezing microtome (SM 2010R, Leica).

**Immunofluorescence staining.** Brain sections were incubated in PBS (pH 7.4) with 10% normal goat serum and 0.1% Triton X-100 (Sigma-Aldrich) for 1 h and then incubated using the following antibody: anti-PV (1:1000, rabbit, Swant, PV 27) (overnight, at 4 °C); anti-CTB (1:500, goat, List Labs, Product No. 703) (48 h, at 4 °C). After washing, Alexa Fluor-conjugated secondary antibodies (1:500, goat anti-mouse, Thermos Fisher Scientific A-11001; 1:500, goat anti-rabbit, Thermo Fisher Scientific A-21245). Finally, sections were subsequently mounted onto glass slides for imaging (LSM 780 laser scanning confocal microscope, Carl Zeiss). Image analysis and cell counting were performed using ImageJ software (Fiji, version 1.52p). Optical fiber placements for all mice included in this study are presented in Supplementary Figs. 3, 4a, 5b, 7b, 9b, 10a, e, g, j, 11a, 13a, h, j, 14c, 15a, 19a.

**Sample preparation and ISH procedure for RNAscope.** For Fig. 2a, b, Fresh-frozen brains from adult male C57BL/6NJ mice (8–12 weeks old) were sectioned at a thickness of 16 μm using a Cryostat (Leica Biosystems). Sections were collected onto Superfrost Plus glass slides (Daigger Scientific), immediately placed on dry ice and subsequently transferred to a −80 °C freezer. mRNA signal for *Spp1* and *Ecel1* was detected using the RNAscope fluorescent kit (Advanced Cell Diagnostics). Specifically, glass slides with sections spanning the entire anteroposterior spread of the PVT were removed from the −80 °C freezer, fixed with freshly prepared ice-chilled 4% PFA for 15 min at 4 °C and then dehydrated using a series of ethanol solutions at different concentrations (5 min each, room temperature): 1 × 50%, 1 × 70% and 2 × 100%. Next, sections were treated with Protease IV (Advanced Cell Diagnostics) at room temperature for 30 min. Slides were then washed with PBS twice (1 min each) and dried for 5 min at room temperature, and sections were circled with an ImmEdge Hydrophobic Barrier PAP Pen (Vector Laboratories). Hybridization was performed on a HybEZ oven for 2 h at 40 °C using a *Spp1* or *Ecel1* probe (Advanced Cell Diagnostics). After this, the slides were washed twice with washing buffer (2 min each), then incubated with Hybridize Amp 1-FL for 30 min, Hybridize Amp 2-FL for 15 min, Hybridize Amp 3-FL for 30 min and Hybridize Amp 4-FL for 15 min. Next, the slides were washed twice with washing buffer (2 min each) and coverslips added using Diamond Prolong antifade mounting medium with DAPI (Thermo Fisher Scientific).

**Signal detection and analysis for RNAscope.** Dried slides were examined on an LSM 780 laser scanning confocal microscope (Carl Zeiss) using 20X objective 24 h after the amplification procedure. Signal was subsequently quantified with CellProfiler 3.1.8 using a freely available pipeline (macros) for RNAscope[109]. A protocol with a step-by-step description of how to implement this pipeline for analyzing RNAscope data was recently published[110]. All RNAscope data were analyzed in a blinded manner.

**Whole-cell patch-clamp slice electrophysiology and single-cell slice imaging**
For all electrophysiological and slice imaging experiments, slices were prepared replicating previous methodological procedures[111] and are described herein. Mice were anesthetized with isoflurane and transcardially perfused with an ice-cold NMDG cutting solution (92 mM *N*-Methyl-D-glucamine, 2.5 mM KCl, 1.25 mM $NaH_2PO_4$, 10 mM $MgSO_4$, 0.5 mM $CaCl_2$, 30 mM $NaHCO_3$, 20 mM glucose, 20 mM HEPES, 2 mM thiourea, 5 mM Na-ascorbate, 3 mM Na-pyruvate, at 7.3–7.4 pH gassed with 95% $O_2$ and 5% $CO_2$). Coronal sections (300-μm thick) containing the avTRN or PVT were cut in the ice-cold NMDG cutting solution using

a VT1200S automated vibrating-blade microtome (Leica Biosystems), and were subsequently transferred to a heated incubation chamber containing the NMDG cutting solution at 34–35 °C. After approximately 12 min, slices were transferred to a room temperature (20–24 °C) holding chamber containing a HEPES-modified artificial cerebrospinal fluid (92 nM NaCl, 2.5 mM KCl, 1.25 mM $NaH_2PO_4$, 2 mM $MgSO_4$, 2 mM $CaCl_2$, 30 mM $NaHCO_3$, 25 mM glucose, 20 mM HEPES, 2 mM thiourea, 5 mM Na-ascorbate, 3 mM Na-pyruvate, at 7.3 pH, gassed with 95% $O_2$ and 5% $CO_2$) and remained in the holding chamber until needed. For recordings, slices were transferred to the recording chamber and constantly supplied with a room-temperature ACSF (118 mM NaCl, 2.5 mM KCl, 26.2 mM $NaHCO_3$, 1 mM $NaH_2PO_4$, 20 mM glucose, 2 mM $MgCl_2$, and 2 mM $CaCl_2$, at pH 7.4, gassed with 95% O2 and 5% CO2).

**Whole-cell patch-clamp recordings.** For Fig. 2d, and Supplementary Fig. 2c, d, recordings from avTRN and/or pPVT neurons were obtained with a Multiclamp 700B amplifier (Molecular Devices). Recordings were done under visual guidance using an Olympus BX51 microscope with transmitted light illumination. Recordings were made in ACSF and pharmacological agents were added to the ACSF and bath applied. All recordings were made with borosilicate glass pipettes with tip resistance of 3-6 MΩ. Access resistance was monitored throughout all recordings and recordings where access resistance increased more than 25% or above 30 MΩ were not included in analyses. Optogenetically evoked synaptic responses were achieved by shining a blue LED (470 nm, pE-300^white, CoolLED) over acute slices to drive ChR2-expressing terminals. All recordings were done in voltage-clamp configuration. For all recordings except in Fig. 2e, cells were kept at a holding potential of −70 mV. To assess presynaptic function, a paired-pulse stimulation protocol (50 ms inter-stimulus interval) was used to evoke double-EPSCs, and the paired-pulse ratio (PPR) was quantified as the ratio of the peak amplitude of the second EPSC to that of the first EPSC. For Fig. 2d, e (PL-PVT-NAc), recordings were made using a Cs-based internal solution containing 117 mM Cs methanesulfonate, 10 mM HEPES, 2.5 mM $MgCl_2$, 2 mM $Na_2$-ATP, 0.4 mM $Na_2$-GTP, 10 mM $Na_2$-phosphocreatine, 0.6 mM EGTA, 5 mM QX-314 at pH 7.2 and 288-290 mOSM. For Fig. 2d (TRN-PVT-NAc), Supplementary Fig. 2c, d recordings were made using a Cs-based internal solution containing 130 mM CsCl, 10 mM HEPES, 1 mM $MgCl_2$, 4 mM NaCl, 4 mM Mg-ATP, 0.3 mM $Na_2$-GTP, 10 mM $Na_2$-phosphocreatine, 0.1 mM EGTA, 5 mM QX-314 at pH 7.3 and 288-290 mOSM. Retrogradely-labeled cells (CTB 555) were identified bases on their red fluorescence using a green LED (pE-300^white, CoolLED). To isolate monosynaptic responses, all recordings were done in the presence of TTX and 4AP. Electrophysiological recordings were collected using pClamp 10 (Molecular Devices, CA).

**Dual opsin stimulation experiments.** For Fig. 2e, two distinct excitatory opsins were selectively expressed in PL and TRN. Opsins for each target were counterbalanced across subjects. For 5 mice (8 slices) AAV2-syn-FLEX-ChrimsonR-tdTomato was injected in the TRN and AAV5-CaMKII-ChR2(H134R)-EYFP was injected into in the PL. For 2 mice (4 slices) AAV9-syn-ChrimsonR-tdTomato was injected into the PL and AAV2-Ef1α-DIO-hChR2(H134R)-EYFP was injected into the TRN. Recordings were done in voltage clamp mode and using Cs-based internal solution as described above, with TTX and 4AP present in the bath. The holding potential was varied between −70 mV or 0 mV to isolate PL-mediated EPSCs and TRN-mediated IPSCs respectively. A blue light was used to excite ChR2 and a fiber optic cable from a red laser was attached to the recording chamber (630 nm, Opto Engine LLC) to excite ChrimsonR. Given that red-shifted opsins like ChrimsonR can be excited by blue-shifted light (i.e., 470 nm)[112], and that the two inputs onto PVT neurons were either excitatory or inhibitory, we implemented a protocol that minimized potential contamination of

postsynaptic currents driven by blue light activation of ChrimsonR. First, cells were held at a membrane potential necessary to isolate the postsynaptic current (PSCs) dependent on the location of ChrimsonR expression (e.g., 0 mV for ChrimsonR expression in TRN). After a baseline recording was established, the appropriate antagonists were bath applied to block the PSCs (e.g., PTX). Once the current was reliably blocked, the holding potential was changed to isolate the PSC driven by the neurons expressing ChR2 (e.g., −70 mV for ChR2 expression in PL).

**Single-cell Ca2± and GABASnFR recordings in brain slice.** For Fig. 8l–n and Supplementary. Fig. 18, acute brain slices containing the PVT were placed in the recording chamber containing ACSF and remained there until imaging finished. Images were obtained using fluorescence microscope (Olympus BX51 microscope) with an Orca Flash 4.0 LT camera and HCImage Live (Hamamatsu Corporation, PA) software. A LED (Lumen 300-LED, Prior Scientific) was used to excite jGCaMP7s. For optogenetically evoked rebound imaging using ChrimsonR, optogenetic light stimulation was achieved using a red laser (635 nm) that delivered pulses at 20 Hz for 2 s via an optical fiber aimed at the slice. For validation of GABASnFR, 10 μM of GABA was bath applied. Images obtained before light stimulation or drug application served as baseline, and the fluorescence changes of jGCaMP8f after light stimulation or drug application were analyzed with ImageJ software. To isolate the GABAergic effects on GCaMP rebound responses, initial recordings were made in ACSF to establish a baseline rebound response, then selective antagonists PTX (50 μM) and CGP 55845 (5 μM) were bath applied.

**Pharmacological agents.** Tetrodotoxin (TTX, Cat. No. 1078), 4-aminopyridine (4AP, Cat. No. 0940), 2,3-Dioxo-6-nitro-1,2,3,4-tetra-hydrobenzo[f]quinoxaline-7-sulfonamide (NBQX, Cat. No. 0373), D-2-amino-5-phosphovalerate (APV, Cat. No 0106), picrotoxin (PTX, Cat. No. 1128) and CGP 55845 hydrochloride (CGP, Cat. No. 1248) were obtained from Tocris Bio-Techne, all other salts were obtained from MilliporeSigma.

**Statistical analysis and reproducibility**
All data were plotted and analyzed with OriginPro version 2016 and version 2018 (OriginLab) and GraphPad Prism (version 8.0.1, GraphPad Software). Electrophysiological data was analyzed using Igor Pro 9 (WaveMetrics, OR). All data are presented as mean ± s.e.m. There were no assumptions or corrections made before data analysis. We conducted a thorough assessment of normality in a pilot trial-based fiber photometry dataset (Fig. 2d–f and Fig. 4), which led us to conclude that the data adhered to a normal distribution. This assessment formed the basis for our decision to assume the normality of the data throughout the entirety of our study, ensuring the appropriateness of subsequent parametric statistical analyses. Differences between two groups were tested with a two-tailed Student's $t$ test; differences among multiple groups were examined with analysis of variance (ANOVA, one-way and two-way repeated-measures; Mixed-effects model, restricted maximum likelihood (REML); group comparisons) corrected by two-stage linear step-up procedure of Benjamini, Krieger and Yekutieli; $P < 0.05$ was considered significant (See Fiber photometry experiments in Source Data). The sample sizes used in our study, such as the numbers of animals, are typically the same or exceed those estimated by power analysis (power = 0.80, α = 0.05). For tracing experiments, the sample size is 2–5 mice. For fiber photometry experiments alone, the sample size is 4–6 mice. For fiber photometry experiments with optogenetic and/or chemogenetic manipulations, the sample size is 3–4 mice. For optogenetic behavior experiments, the sample size is 6–13 mice. For ex vivo electrophysiology experiments, the sample size was 3–8 mice and 1–3 slices per mouse. All experiments were replicated at least once, and similar results were obtained. All representative images were

repeated at least once, and similar results were obtained. All experiments were randomized. For all behavior experiments, investigators were blinded to allocation during experiments. Data distribution was assumed to be normal, but this was not formally tested.

**Reporting summary**
Further information on research design is available in the Nature Portfolio Reporting Summary linked to this article.

## Data availability
All the data that support the findings presented in this study are available from the corresponding author upon request. Source data are provided with this paper.

## Code availability
R code used to analyze active avoidance behavior and photometric signal is available at the following repository: https://github.com/Penzolab/Data-analysis-of-Two-way-active-avoidance-task.git. https://doi.org/10.5281/zenodo.12707790

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

## Acknowledgements

The authors thank the members of the Section on the Neural Circuits of Emotion and Motivation of NIMH who offered scientific feedback. In addition, we thank the NIMH IRP Rodent Behavioral Core for their support with the development of the custom apparatus for the 2AA task, and the NIMH IRP Systems Neuroscience Imaging Resource and the NICHD Microscopy and Imaging Core for assistance with microscopy tools used for histological assessment. Lastly, we thank Drs. Fabricio Do Monte, Michael Halassa, and Briana Carroll for offering scientific and writing feedback. This work was supported by the NIMH Intramural Research Program (1ZIAMH002950, to M.A.P.) and NIGMS Postdoctoral Research and Training Award (1FI2GM146653-01 to J.J.O.).

## Author contributions

J.Ma and J.J.O'Malley performed all experiments. M.Kreiker and M.Kindel assisted with anatomical studies. Y.Leng assisted with RNAscope experiments. M.Kreiker, Y.Leng, and I.Khan assisted with histological procedures. J.Ma, J.J.O'Malley, and M.Kreiker analyzed the data. J.Ma, J.J.O'Malley, and M.A.Penzo designed the study, interpreted results, and wrote the paper.

## Funding

## Competing interests

The authors declare no competing interests.
