## [Peer Review File · Nature Communications]

REVIEWERS' COMMENTS

Reviewer #1 (Remarks to the Author):

The authors addressed my remaining concern by adding a circuit diagram (Extended Figure 12).

Below my comments on the critiques of the third reviewer:

Major Points:

1) Layer 5 vs 6 specificity for direct vs indirect pathway:

The addition of mapping monosynaptic inputs to PVT-NAc using the cTRIO rabies virus methods provides strong evidence of layer 6 specificity of the direct projections. I agree with the reviewer and the authors that layer 6 could still contribute to the indirect projections. Tuning down the layer specificity of the direct/indirect pathway model in the discussion, as provided in the revised manuscript, addresses this concern.

2) Layer 6 inputs to PVT are modest, yet optogenetic perturbation has strong behavioral effects:

I agree with the authors that a “weak” modulator pathway still can be able to regulate/modulate behavior

3) 3A) Strong feed-forward inhibition is not specific to this PrL cortico-thalamic pathway or to L5 but mediated by L6 in visual cortex: This is the same discussion as for point 1.

3B) Omission of Dong et al. 2019 Nat Neuro:

This is an interesting study showing a similar mechanism but in a different thalamic sub-circuit and under different behavioral conditions. I am glad reviewer 3 brought this study to the authors' attention, so that it can be cited now (ref. 107).

4) The variables for all statistic are still individual trials not the average response of individual animals:

The revised analysis using mixed-effect models takes care of this criticism.

5) The authors still fail to provide conclusive evidence for a rebound response in PVT as a response to TRN inhibition thus the true role of the ffw inhibition via the TRN remains obscure:

The data are consistent with a rebound response but are not fully conclusive. Personally, for me providing T-Type Ca^{2+} channel dependency of this mechanism during behavior goes beyond the scope of the manuscript.

Summary:

Ma and O'Malley et al present exciting work describing a paradoxical collection of observations relating Ca²⁺ dynamics in the prelimbic (PL) cortex to periventricular thalamic (PVT) circuit during a learned active avoidance behavior. Through a large set of circuit mapping (anatomical and electrophysiological), optogenetic, and fiber photometry experiments, the authors present a novel framework for corticothalamic interactions involving the dorsal midline thalamus that is distinct from corticothalamic interactions reported previously in sensory domains. This work builds on past publications by this group demonstrating a causal role of the PVT-NAc pathway in active avoidance decisions (Ma et al., 2020). The central idea in the current study is that PVT-NAc activity is transiently inhibited and subsequently undergoes post-inhibitory rebound activity in response to a pull-release mechanism involving transient feedforward inhibition in the PL-avTRN-PVT pathway. The conclusion that the PL-avTRN-PVT pathway dynamics are causally involved and upstream of the previously described PVT-NAc circuit mechanism is supported by bidirectional optogenetics experiments that specifically perturb either PL projections to the avTRN or avTRN projections to the PVT. The authors go to great lengths to contextualize the circuit elements and their dynamics through a multipronged approach including viral circuit mapping, synaptic electrophysiology, and *in vivo* fiber photometry with and without simultaneous optogenetic manipulations. The experiments and their conclusions will be of broad interest to the readership of *Nature Communications*. The authors have added important language to acknowledge some of the limitations of their dataset. Ultimately, there is one additional point of analysis I would like to see as well as rectification of a few issues regarding data presentation and clarity of the statistical analyses performed, which I have outline below.

2 overall suggestions:

I would suggest using a different color scheme for either the heatmaps or the avoidance vs escape trial plots. Coloring both types of data blue and red requires extra effort from readers to not make associations between the data. Secondly, in future manuscript submissions, please do some work to make it easier for reviewers to connect text with associated figures. There are many ways to achieve this (e.g. use in-line figures, include figure numbers (minimum!) and legends on the same page as the figure panels, or make a live document with hyperlinks to the figures placed within the text).

1 Requested Point of Clarification

For the transient optogenetic perturbations occurring at WS onset (Figures 5F-G, 6F-G), I would like to know why there is not a statistically significant change in the 'Normalized Latency in Avoidance' if the dynamics the authors highlight are actually responsible for 'driving' the decision to avoid. One would expect that optogenetically driving an earlier occurrence of the decision signal at WS onset would cause a shortening of the latency to avoid, but this is not seen for either PL-avTRN or avTRN-PVT transient manipulations. Moreover, in the case of long lasting optogenetic excitation of PL-avTRN that would prevent the pull-release inhibitory rebound, the avoidance responses (though

less frequent) happen faster (Figure 5H-I). These results are difficult to reconcile with the simple model proposed.

Issues with Figures:

Figure 5C (pasted below), is the dotted black line in the avoidance AUC plot supposed to be set at 0?

Figure 5D (pasted to the right), thick black line at bottom is difficult to interpret perhaps due to mis-alignment (inconsistent with placement of same indicator in Figure 4D) with the dotted vertical line and the white space that breaks the black thick line at about 1.5 seconds along the x-axis. The figure legend and methods text lead me to understand the analysis as comparisons between trial types being performed at each 1 second timepoint. If this is correct, consider shifting the black thick lines to a more intuitive location.

Figure 4D, 5D: Perhaps this is a minor point, but I find it very confusing and hard to believe the statistics described by this line “Blue or red thick lines along x axis indicate where post hoc tests found the first significant change from the previous bin within trial types ($P < 0.05$, Mixed-effects model (REML))”. Several things make me skeptical. First example: 4D right upper blue AUC data (pasted below to the left), are we really supposed to believe that the -1 second time bin is significantly different from the -2 second time bin as indicated by the thick blue line? I have similar questions about 5C red lower AUC data (pasted to the right). What is the primary point being made by these figures? What is the significance of the within trial type comparisons and the

different time points at which the different trial types reach statistical significance? I raised this issue in the last reviewer report, but the changes made by the authors did not resolve the issue. If there’s an important message conveyed by these plots, I am totally missing it.

Summary:

Ma and O'Malley et al present exciting work describing a paradoxical collection of observations relating Ca²⁺ dynamics in the prelimbic (PL) cortex to periventricular thalamic (PVT) circuit during a learned active avoidance behavior. Through a large set of circuit mapping (anatomical and electrophysiological), optogenetic, and fiber photometry experiments, the authors present a novel framework for corticothalamic interactions involving the dorsal midline thalamus that is distinct from corticothalamic interactions reported previously in sensory domains. This work builds on past publications by this group demonstrating a causal role of the PVT-NAc pathway in active avoidance decisions (Ma et al., 2020). The central idea in the current study is that PVT-NAc activity is transiently inhibited and subsequently undergoes post-inhibitory rebound activity in response to a pull-release mechanism involving transient feedforward inhibition in the PL-avTRN-PVT pathway. The conclusion that the PL-avTRN-PVT pathway dynamics are causally involved and upstream of the previously described PVT-NAc circuit mechanism is supported by bidirectional optogenetics experiments that specifically perturb either PL projections to the avTRN or avTRN projections to the PVT. The authors go to great lengths to contextualize the circuit elements and their dynamics through a multipronged approach including viral circuit mapping, synaptic electrophysiology, and *in vivo* fiber photometry with and without simultaneous optogenetic manipulations. The experiments and their conclusions will be of broad interest to the readership of *Nature Communications*. The authors have added important language to acknowledge some of the limitations of their dataset. Ultimately, there is one additional point of analysis I would like to see as well as rectification of a few issues regarding data presentation and clarity of the statistical analyses performed, which I have outline below.

Overall suggestions:

I would suggest using a different color scheme for either the heatmaps or the avoidance vs escape trial plots. Coloring both types of data blue and red requires extra effort from readers to not make associations between the data. Secondly, in future manuscript submissions, please do some work to make it easier for reviewers to connect text with associated figures. There are many ways to achieve this (e.g. use in-line figures, include figure numbers (minimum!) and legends on the same page as the figure panels, or make a live document with hyperlinks to the figures placed within the text).

We thank the reviewer for these useful suggestions, which we will keep in mind for future submissions. Here, we have changed the color scheme for all heatmaps to eliminate potential confounds between these and trial type-based average responses (i.e., Avoidances and Escape).

Requested Point of Clarification

For the transient optogenetic perturbations occurring at WS onset (Figures 5F-G, 6F-G), I would like to know why there is not a statistically significant change in the 'Normalized Latency in Avoidance' if the dynamics the authors highlight are actually responsible for 'driving' the decision to avoid. One would expect that optogenetically driving an earlier occurrence of the decision signal at WS onset would cause a shortening of the latency to avoid, but this is not seen for either PL-avTRN or avTRN-PVT transient manipulations. Moreover, in the case of long lasting optogenetic excitation of PL-avTRN that would prevent the pull-release inhibitory rebound, the avoidance responses (though

less frequent) happen faster (Figure 5H-I). These results are difficult to reconcile with the simple model proposed.

The reviewer makes an important point, and we offer the following explanation that we hope clarifies the findings and have incorporated into our discussion. Our findings show that both PL and TRN become briefly engaged and subsequently suppressed during avoidance behaviors. Our current model is that this dynamic response is part of two processes necessary for the expression of the behavior. The first is the initial decision to avoid which does not require immediate action, and second is the “motivational” drive to engage in the behavior. The initial engagement of the indirect pathway primes the PVT (and mouse) for avoidance, that is the decision to avoid has been made but the still requires a driver for the action. We think this is where the suppression of the pathway becomes critical for the onset of the behavior. If this is true, then brief engagement of the pathway without suppressing it should enhance avoidance but not affect aspects of the behavior occurrence like latency to avoid. Likewise, if the suppression of the pathway following transient activation at WS onset is the driver, then inhibiting it shortly after WS response should precipitate avoidance occurrence (i.e., decrease latency). Testing this prediction from our data would require a close-loop approach whereby increased activity in the PL-TRN pathway at WS onset will drive optogenetic suppression of this pathway. This is an interesting approach that requires some effort to implement, and we would consider implementing in a follow up study.

The observation that, despite significantly decreasing the Avoidance Rate, sustained excitation of the PL-TRN pathway during the WS results in a drastic reduction in the Latency to Avoid is indeed intriguing, as noted by the reviewer. One potential interpretation of this result is that while sustained stimulation of the PL-TRN pathway reduces the likelihood of engaging in avoidance behavior, as predicted by our model, this manipulation may affect PL projections to extra-thalamic areas (e.g., collaterals to the ventral striatum and/or the ventral tegmental area), which may impact motivational processes.

Issues with Figures:

Figure 5C (pasted below), is the dotted black line in the avoidance AUC plot supposed to be set at 0?

We thank the reviewer for noting this. We have corrected this in the figure.

Figure 5D (pasted to the right), thick black line at bottom is difficult to interpret perhaps due to mis-alignment (inconsistent with placement of same indicator in Figure 4D) with the dotted vertical line and the white space that breaks the black thick line at about 1.5 seconds along the x-axis. The figure legend and methods text lead me to understand the analysis as comparisons between trial types being performed at each 1 second timepoint. If this is correct, consider shifting the black thick lines to a more intuitive location.

Figure 4D, 5D: Perhaps this is a minor point, but I find it very confusing and hard to believe the statistics described by this line “Blue or red thick lines along x axis indicate where post hoc tests found the first significant change from the previous bin within trial types ($P < 0.05$, Mixed-effects model (REML)”. Several things make me skeptical. First example: 4D right upper blue AUC data (pasted below to the left), are we really supposed to believe that the -1 second time bin is significantly different from the -2 second time bin as indicated by the thick blue line? I have similar questions about 5C red lower AUC data (pasted to the right). What is the primary point being made by these figures? What is the significance of the within trial type comparisons and the

different time points at which the different trial types reach statistical significance? I raised this issue in the last reviewer report, but the changes made by the authors did not resolve the issue. If there's an important message conveyed by these plots, I am totally missing it.

We appreciate the reviewer for these concerns and have reworded the phrasing in our figure legends to hopefully clarify these points with the following:

“For all quantifications, multiple comparisons were conducted corrected by a two-stage linear step-up procedure, black lines along the x-axis indicate significant changes reported between groups, and red or blue lines denote the first significant change from the previous bin for within trial type comparisons.”

The black lines show between trial type significance moment to moment during the recording (sampled at 8 Hz, not binned in 1s bins). This is done to highlight the significant differences found between trial types without overwhelming the figure with notations. Additionally, because this is a moment-to-moment comparison from data sampled at 8 Hz and not binned in 1 s the black bar shown here from Fig 6d does come just before the event as there is a significant difference prior to shuttle initiation between escape trials and avoidance trials.

The red or blue lines are for within group comparisons only for the AUC 1s binned data. The purpose of this is to highlight differences within a group (Escape vs Avoidance or Light ON vs Light OFF) and to show where there is a significant change in the signal but to prevent comparisons between groups when there is no significance there. We chose the first point that has significance as we think this is the most meaningful point to highlight, as that is when there is the first difference in activity, and having the single bar helps to denote where the that significant change is relative to any event of interest. We understand the reviewers point about what does this mean and why is there a significant difference. For Fig 4d the most parsimonious expiation is that there is little error, and the bin capture part of the point where the signal begins to decrease. For the -2s bin for Escape trials in Fig 6c, there is a significant effect but we do not make any interpretations about this specific point to be agnostic and to that point want to maintain transparency and highlight events that the statistical tests find significant, and all data are

available on the source data file.

We hope this further helps to clarify the issue brought up by the reviewer.